# Loss of centromere function drives karyotype evolution in closely related *Malassezia* species

**Sundar Ram Sankaranarayanan[1], Giuseppe Ianiri[2†], Marco A Coelho[2], Md Hashim Reza[1], Bhagya C Thimmappa[1‡], Promit Ganguly[1], Rakesh Netha Vadnala[3], Sheng Sun[2], Rahul Siddharthan[3], Christian Tellgren-Roth[4], Thomas L Dawson Jnr[5,6], Joseph Heitman[2\*], Kaustuv Sanyal[1\*]**

[1]Molecular Mycology Laboratory, Molecular Biology and Genetics Unit, Jawaharlal Nehru Centre for Advanced Scientific Research, Bengaluru, India; [2]Department of Molecular Genetics and Microbiology, Duke University Medical Center, Durham, United States; [3]The Institute of Mathematical Sciences/HBNI, Chennai, India; [4]National Genomics Infrastructure, Science for Life Laboratory, Department of Immunology, Genetics and Pathology, Uppsala University, Uppsala, Sweden; [5]Skin Research Institute Singapore, Agency for Science, Technology and Research (A\*STAR), Singapore, Singapore; [6]Department of Drug Discovery, Medical University of South Carolina, School of Pharmacy, Charleston, United States

**\*For correspondence:**
heitm001@duke.edu (JH);
sanyal@jncasr.ac.in (KS)

**Present address:** [†]Department of Agricultural, Environmental and Food Sciences, University of Molise, Campobasso, Italy; [‡]Department of Biochemistry, Robert-Cedergren Centre for Bioinformatics and Genomics, University of Montreal, Montreal, Canada

**Competing interests:** The authors declare that no competing interests exist.

**Abstract** Genomic rearrangements associated with speciation often result in variation in chromosome number among closely related species. *Malassezia* species show variable karyotypes ranging between six and nine chromosomes. Here, we experimentally identified all eight centromeres in *M. sympodialis* as 3–5-kb long kinetochore-bound regions that span an AT-rich core and are depleted of the canonical histone H3. Centromeres of similar sequence features were identified as CENP-A-rich regions in *Malassezia furfur*, which has seven chromosomes, and histone H3 depleted regions in *Malassezia slooffiae* and *Malassezia globosa* with nine chromosomes each. Analysis of synteny conservation across centromeres with newly generated chromosome-level genome assemblies suggests two distinct mechanisms of chromosome number reduction from an inferred nine-chromosome ancestral state: (a) chromosome breakage followed by loss of centromere DNA and (b) centromere inactivation accompanied by changes in DNA sequence following chromosome–chromosome fusion. We propose that AT-rich centromeres drive karyotype diversity in the *Malassezia* species complex through breakage and inactivation.

## Introduction

Centromeres are the genomic loci on which the kinetochore, a multi-subunit complex, assembles to facilitate high-fidelity chromosome segregation. The centromere-specific histone H3 variant CENP-A is the epigenetic hallmark of centromeres, as it replaces canonical histone H3 in the nucleosomes to make specialized centromeric chromatin that acts as the foundation to recruit other kinetochore proteins. A remarkable diversity in the organization of centromere DNA sequences has been observed to accomplish this conserved role (*Roy and Sanyal, 2011*; *Yadav et al., 2018b*).

The smallest known centromeres are the point centromeres present in budding yeasts of the family Saccharomycetaceae that span <200 bp in length (*Clarke and Carbon, 1980*; *Gordon et al., 2011*; *Kobayashi et al., 2015*). These centromeres are organized into conserved DNA elements I, II, and III that are recognized by a cognate kinetochore protein complex called the CBF3 complex,

**eLife digest** Millions of yeast, bacteria and other microbes live in or on the human body. A type of yeast known as *Malassezia* is one of the most abundant microbes living on our skin. Generally, *Malassezia* do not cause symptoms in humans but are associated with dandruff, dermatitis and other skin conditions in susceptible individuals. They have also been found in the human gut, where they exacerbate Crohn's disease and pancreatic cancer.

There are 18 closely related species of *Malassezia* and all have an unusually small amount of genetic material compared with other types of yeast. In yeast, like in humans, the genetic material is divided among several chromosomes. The number of chromosomes in different *Malassezia* species varies between six and nine.

A region of each chromosome known as the centromere is responsible for ensuring that the equal numbers of chromosomes are passed on to their offspring. This means that any defects in centromeres can lead to the daughter yeast cells inheriting unequal numbers of chromosomes. Changes in chromosome number can drive the evolution of new species, but it remains unclear if and how centromere loss may have contributed to the evolution of *Malassezia* species.

Sankaranarayanan et al. have now used biochemical, molecular genetic, and comparative genomic approaches to study the chromosomes of *Malassezia* species. The experiments revealed that nine *Malassezia* species had centromeres that shared common features such as being rich in adenine and thymine nucleotides, two of the building blocks of DNA.

Sankaranarayanan et al. propose that these adenines and thymines make the centromeres more fragile leading to occasional breaks. This may have contributed to the loss of centromeres in some *Malassezia* cells and helped new species to evolve with fewer chromosomes.

A better understanding of how *Malassezia* organize their genetic material should enable in-depth studies of how these yeasts interact with their human hosts and how they contribute to skin disease, cancer, Crohn's disease and other health conditions. More broadly, these findings may help scientists to better understand how changes in chromosomes cause new species to evolve.

making them genetically defined centromeres. Small regional centromeres, identified in several *Candida* species, form the second category (*Sanyal et al., 2004*; *Padmanabhan et al., 2008*; *Kapoor et al., 2015*; *Chatterjee et al., 2016*) and have a 2–5-kb region enriched by kinetochore proteins. These centromeres can either have unique DNA sequences or a homogenized core that is flanked by inverted repeats. The third type of centromere structure is the large regional centromere, which is often repetitive in sequence and spans more than 15 kb. Large regional centromeres can be transposon-enriched, as in *Cryptococcus* species, or organized into repeat structures around a central core, as in *Schizosaccharomyces pombe* (*Chikashige et al., 1989*; *Clarke and Baum, 1990*; *Sun et al., 2017*; *Yadav et al., 2018b*).

Although the organization of DNA elements is variable, a majority of known centromeres share AT-richness as a common feature. Examples include the CDEII of point centromeres, central core sequences in *S. pombe*, and centromeres of *Neurospora crassa*, *Magnaporthe oryzae*, *Plasmodium falciparum*, and diatoms (*Fitzgerald-Hayes et al., 1982*; *Iwanaga et al., 2010*; *Rhind et al., 2011*; *Kapoor et al., 2015*; *Diner et al., 2017*; *Yadav et al., 2019*). Even the recently described mosaic centromere structure observed in *Mucor circinelloides* that has lost CENP-A comprises an AT-rich kinetochore-bound core region (*Navarro-Mendoza et al., 2019*). Although suppression of recombination around centromeres has been correlated with reduced GC content (*Lynch et al., 2010*), the genetic underpinning that determines how an AT-rich DNA region favors kinetochore assembly remains unclear. Ironically, AT-rich sequences have been shown to be fragile sites within a chromosome (*Zhang and Freudenreich, 2007*).

Several lines of evidence suggest that centromeres are species-specific and are among the most rapidly evolving genomic regions, showing variation even between closely related species (*Bensasson et al., 2008*; *Padmanabhan et al., 2008*; *Rhind et al., 2011*; *Roy and Sanyal, 2011*). This evolution is accompanied by the concomitant evolution of CENP-A and the associated kinetochore proteins (*Talbert et al., 2004*). Functional incompatibilities between centromeres result in uniparental genome elimination in interspecies hybrids (*Ravi and Chan, 2010*; *Sanei et al., 2011*). The

divergent nature of centromeres is proposed to be a driving force for speciation (*Henikoff et al., 2001*; *Malik and Henikoff, 2009*).

Asexual organisms, by virtue of inter- and intra-chromosomal rearrangements, diversify into species clusters that are distinct in genotype and morphology (*Barraclough et al., 2003*). These genotypic differences include changes in both chromosomal organization and number. Centromere function is directly related to karyotype stabilization following a change in chromosome number. Rearrangements in the form of telomere–telomere fusions and nested chromosome insertions (NCIs), wherein an entire donor chromosome is 'inserted' into or near the centromere of a non-homologous recipient chromosome, are among the major sources of chromosome number reduction (*Lysak, 2014*). Such events often result in the formation of dicentric chromosomes that are subsequently stabilized by breakage-fusion-bridge cycles (*McClintock, 1941*) or via inactivation of one centromere through different mechanisms (*Han et al., 2009*; *Sato et al., 2012*). Well-known examples of telomere–telomere fusions include the formation of extant human chromosome 2 by fusion of two ancestral chromosomes (*Yunis and Prakash, 1982*; *IJdo et al., 1991*), the reduction in karyotype observed within members of the Saccharomycotina such as *Candida glabrata*, *Vanderwaltozyma polyspora*, *Kluyveromyces lactis,* and *Zygosaccharomyces rouxii* (*Gordon et al., 2011*), and the exchange of chromosomal arms seen in plants and fungi (*Schubert and Lysak, 2011*; *Sun et al., 2017*). NCIs have predominantly shaped karyotype evolution in grasses (*Murat et al., 2010*). Reduction of chromosome number by centromere loss has also been reported (*Gordon et al., 2011*).

To investigate whether centromere breakage can be a natural source of karyotype diversity in closely related species, we sought to identify centromeres in a group of *Malassezia* yeast species that exhibit variation in chromosome number. *Malassezia* species are lipid-dependent basidiomycetous fungi that are naturally found as part of the animal skin microbiome (*Theelen et al., 2018*). At present, the *Malassezia* genus includes 18 species divided into three clades: A, B, and C. These species also have unusually compact genomes of less than 9 Mb, organized into six to nine chromosomes as revealed by electrophoretic karyotyping of some of these species (*Boekhout and Bosboom, 1994*; *Boekhout et al., 1998*; *Wu et al., 2015*). Fungemia-associated species such as *Malassezia furfur* belong to Clade A. Clade B includes common inhabitants of human skin that are phylogenetically clustered into two subgroups, namely Clade B1 that contains *Malassezia globosa* and *Malassezia restricta* and Clade B2 that contains *Malassezia sympodialis* and related species. Clade C includes *Malassezia slooffiae* and *Malassezia cuniculi*, which diverged earlier from a *Malassezia* common ancestor (*Wu et al., 2015*; *Lorch et al., 2018*).

Besides humans, *Malassezia* species have been detected on the skin of other animals. For example, *M. slooffiae* was isolated from cows and goats, *M. equina* from horses, *M. brasiliensis* and *M. psittaci* from parrots, and the cold-tolerant species *M. vespertilionis* from bats (*Lorch et al., 2018*; *Theelen et al., 2018*). In addition, culture-independent studies of fungi from environmental samples showed that *Malassezia* species that are closely related to those found on human skin were also detected in diverse niches, such as deep-sea vents, soil invertebrates, hydrothermal vents, corals, and Antarctic soils (*Amend, 2014*). More than ten *Malassezia* species have been detected as a part of the human skin microbiome (*Findley and Grice, 2014*). The human skin commensals such as *M. globosa, M. restricta,* and *M. sympodialis* have been associated with dermatological conditions such as dandruff/seborrheic dermatitis, atopic dermatitis, and folliculitis (*Theelen et al., 2018*). Recent reports implicate *M. restricta* in conditions such as Crohn's disease and *M. globosa* in the progression of pancreatic cancer (pancreatic ductal adenocarcinoma) (*Aykut et al., 2019*; *Limon et al., 2019*). Elevated levels of *Malassezia* species and the resulting inflammatory host response have been implicated in both of these disease states. The nature of genomic rearrangements in each species may influence its ability to adapt and cause disease in a specific host niche. Thus, studying the mechanisms of karyotype evolution is an important step towards understanding the evolution of the *Malassezia* species complex.

Kinetochore proteins serve as useful tools in the identification of centromeres because of their centromere-exclusive localization. CENP-A replaces histone H3 in the centromeric nucleosomes. This has been shown as a reduction in histone H3 levels at the centromeres in *Candida lusitaniae* (*Kapoor et al., 2015*) and in a human neocentromere (*Lo et al., 2001*). These CENP-A nucleosomes act as a foundation to recruit CENP-C, the KMN (KNL1C-MIS12C-NDC80C) network, and other kinetochore proteins (*Musacchio and Desai, 2017*).

In this study, we experimentally validated all of the eight centromeres of *M. sympodialis* using the Mtw1 protein (Mis12 in *S. pombe*), a subunit of the KMN complex, as the kinetochore marker. The Mis12 complex proteins are evolutionarily conserved outer kinetochore proteins that link the chromatin-associated inner kinetochore proteins to the microtubule-associated outer kinetochore proteins. Members of the Mis12 complex localize to centromeres in other organisms (*Goshima et al., 1999*; *Goshima et al., 2003*; *Westermann et al., 2003*; *Roy et al., 2011*). Recent studies suggest that the protein domains associated with the Mis12 complex members are exclusive to kinetochore proteins and are not detected in any other proteins, making them attractive tools for identifying centromere sequences (*Tromer et al., 2019*). Using the features of centromeres of *M. sympodialis* and newly generated chromosome-level genome assemblies, we predicted centromeres in related *Malassezia* species carrying seven, eight, or nine chromosomes, and experimentally validated the centromere identity in representative species of each karyotype, each belonging to a different *Malassezia* clade. We employed gene synteny conservation across these centromeres to understand their transitions from an inferred ancestral state of nine chromosomes. On the basis of our results, we propose that centromere loss by two distinct mechanisms drives karyotype diversity.

## Results

### Chromosome number varies in the *Malassezia* species complex

Previous reports that are based on pulsed-field gel electrophoresis (PFGE) have suggested that chromosome number varies within the *Malassezia* species complex. The early diverged species *M. slooffiae* of Clade C was reported to have nine chromosomes (*Boekhout et al., 1998*). Clade B *Malassezia* species are reported to have nine (*M. globosa* and *M. restricta*), eight (*M. sympodialis*), or six chromosomes (*M. pachydermatis*). Among the Clade A species, *M. obtusa* and *M. furfur* CBS14141 were both reported to have seven chromosomes (*Boekhout and Bosboom, 1994*; *Zhu et al., 2017*). A high-quality reference genome is a prerequisite to understanding the rearrangements associated with chromosome number variation. In addition, such a reference genome will also assist in resolving ambiguities in PFGE-based estimates of chromosome number when similar-sized chromosomes are present. Complete genome assemblies were not available for many of the species with reported numbers of chromosomes. To obtain better-assembled reference genomes, we sequenced the genomes of *M. slooffiae* and *M. globosa* as representatives of the nine-chromosome state, and of *M. furfur* as a representative of the seven-chromosome state, using PacBio SMRT sequencing technology (*Figure 1*, *Figure 1—figure supplement 1*).

The *M. globosa* genome was completely assembled into nine contigs with telomeres on both ends (BioSample accession SAMN10720087). We validated these contigs by matching each band on the pulsed-field gel with the contig sizes from the genome assembly, and further confirmed these by chromoblot analysis following PFGE. This analysis shows that chromosome 5 contains the rDNA locus and migrates further than the expected size of 902 kb, as a diffuse ensemble of different sizes along with chromosome 3 (*Figure 1A*). The assembled genome of *M. slooffiae* has 14 contigs of which nine contigs have telomeres on both ends, indicative of nine chromosomes (BioSample accession SAMN10720088). Each of the nine contigs could be assigned to the bands observed in the pulsed-field gel (*Figure 1B*). For *M. furfur*, the final genome assembly consisted of seven contigs with telomeres on both ends and matched the expected chromosome sizes obtained from an earlier PFGE analysis of CBS14141 (*Figure 1C*). The complete genome assembly of *M. sympodialis* reported earlier is distributed into eight chromosomes with telomere repeats on both ends (*Figure 1D*), and serves as a representative of an eight-chromosome state in this study.

Changes in chromosome number are always associated with the birth or loss of centromeres, which stabilizes the karyotype in organisms with monocentric chromosomes. To understand the transitions between these different karyotypic states observed in the *Malassezia* species complex, we sought to validate centromeres in species representative of each karyotype experimentally.

### Kinetochores cluster and localize to the nuclear periphery in *M. sympodialis*

Organisms that have point centromeres possess Ndc10, Cep3, and Ctf13 of the CBF3 complex, a cognate protein complex that is specific to point centromeres. None of these point-centromere-

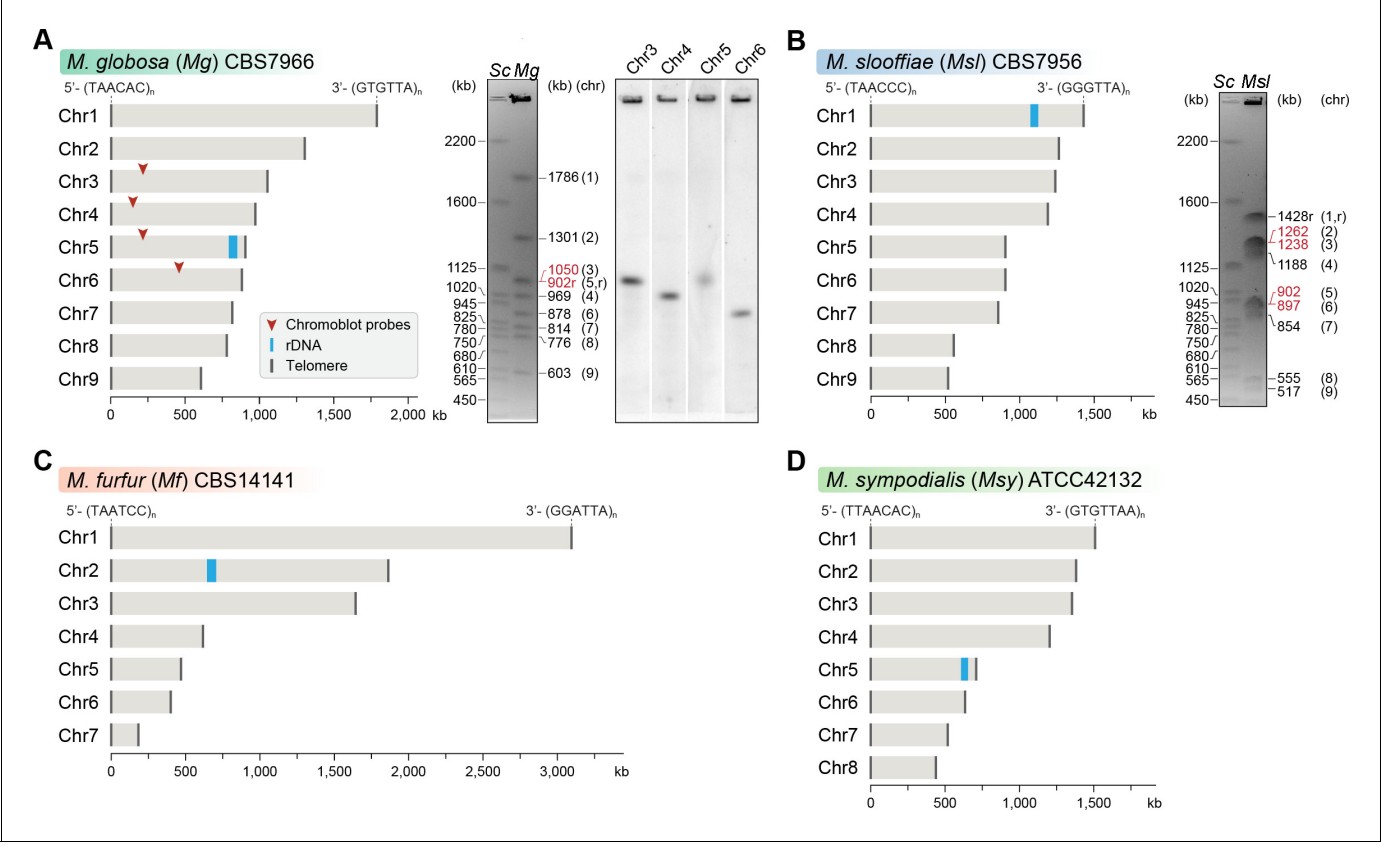

**Figure 1.** Genome assembly and karyotype diversity in representative *Malassezia* species. The genomes of (**A**) *M. globosa*, (**B**) *M. slooffiae*, and (**C**) *M. furfur* were sequenced and assembled in this study, whereas the genome assembly of (**D**) *M. sympodialis* was reported earlier (**Zhu et al., 2017**) and is shown for comparison. In each panel, bar plots represent the assembled chromosomes of the indicated *Malassezia* species, with the telomeres and the ribosomal DNA (rDNA) marked as dark gray and blue bars, respectively. Telomere-repeat motifs are shown at the 5'- and 3'-ends of the Chr1 in each species. Electrophoretic karyotypes of *M. globosa* (*Mg*) and *M. sloffiiae* (*Msl*), are shown in (**A**) and (**B**), respectively, with chromosome sizes estimated from the genome assembly. Chromosomes of *Saccharomyces cerevisiae* (*Sc*) served as size markers. The chromosome containing the rDNA (marked with an 'r'), in *M. globosa,* co-migrates with Chr3. This was assessed by chromoblot hybridization using unique sequences from Chr3, Chr4, Chr5, and Chr6 as probes (regions indicated by red arrowheads). Chromosomes of similar size (denoted in red) migrate together in the gel and appear as a doublet band (i.e. MgChr3–MgChr5, MslChr2–MslChr3, and MslChr5–MslChr6).

The online version of this article includes the following source data and figure supplement(s) for figure 1:

**Figure supplement 1.** Statistics of the genome assemblies of *M. globosa, M. slooffiae,* and *M. furfur* generated in this study.

**Figure supplement 1—source data 1.** Statistics of the genome assemblies of *M. globosa, M. slooffiae,* and *M. furfur* generated in this study.

specific proteins could be detected in *M. sympodialis*. However, we could detect homologs of CENP-A, CENP-C, and most of the outer kinetochore proteins in the genome of *M. sympodialis* (**Figure 2A** and **Figure 2—figure supplement 1**). We functionally expressed an N-terminally GFP-tagged Mtw1 protein (Protein ID: SHO76526.1) from its native promoter, and the expression of the fusion protein was confirmed by western blotting (**Figure 2B**). Upon staining with anti-GFP antibodies and DAPI (4',6-diamidino-2-phenylindole), we were able to detect punctate localization of Mtw1 at the nuclear periphery (**Figure 2C**), consistent with the clustered kinetochore localization observed in other yeasts (**Goshima et al., 1999**; **Euskirchen, 2002**; **Roy et al., 2011**). Live-cell images of MSY001 (GFP-*MTW1*) cells revealed that the kinetochores (GFP-Mtw1) remained clustered throughout the cell cycle, starting from unbudded $G_1$ cells in interphase to large-budded cells in mitosis (**Figure 2D**).

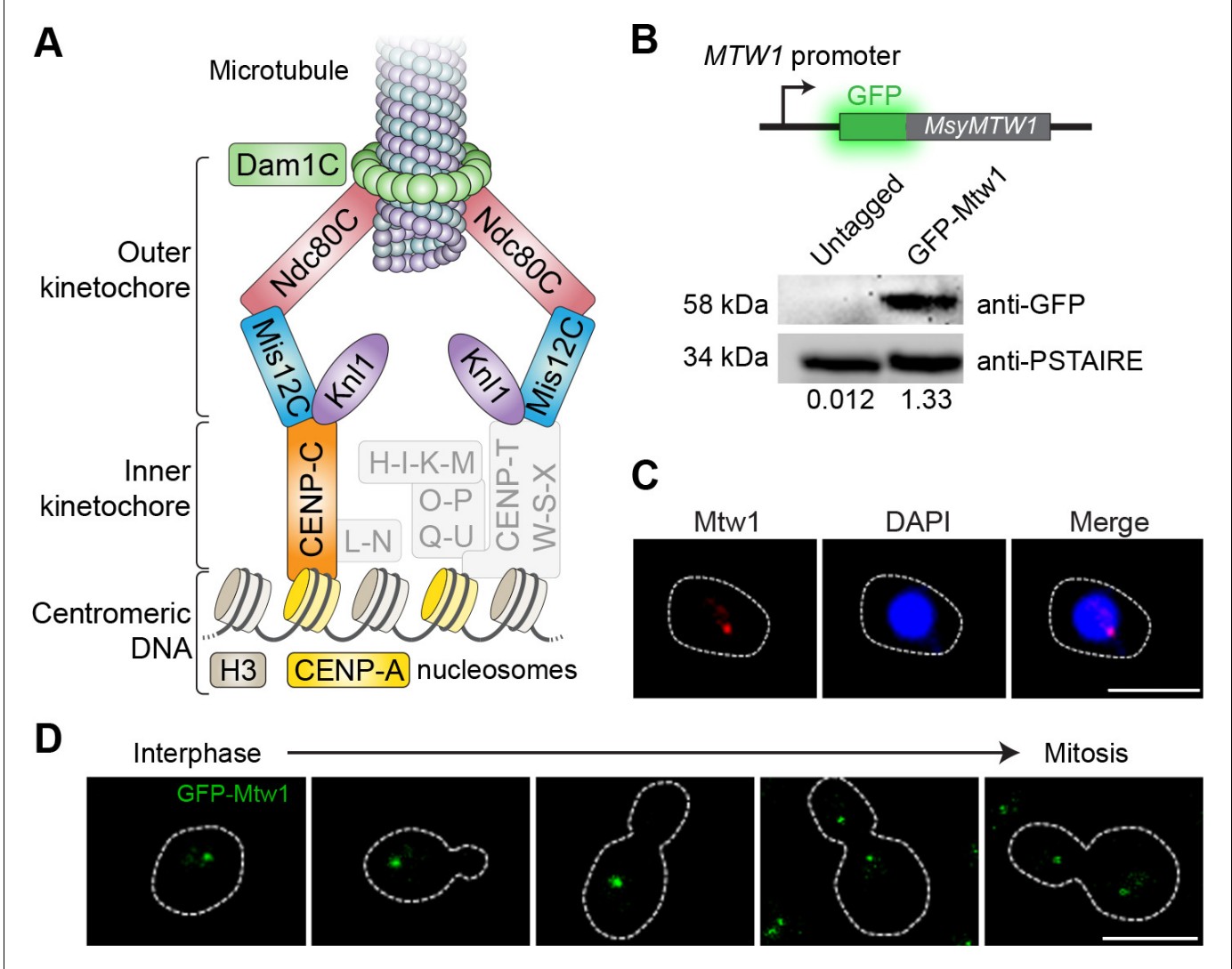

**Figure 2.** Kinetochores cluster and localize at the nuclear periphery in *M. sympodialis*. (A) Schematic of the kinetochore organization of *M. sympodialis*. Gray boxes indicate proteins absent in *M. sympodialis*. The outer kinetochore protein Mtw1 (a component of Mis12C) served as the kinetochore marker in the present study. (B) Line diagram representation of Mtw1 tagged with GFP at the N- terminus. Immunoblot analysis of whole-cell lysates prepared from the untagged *M. sympodialis* strain (ATCC42132) and from GFP-Mtw1 expressing cells (MSY001) probed with anti-GFP antibodies and anti-PSTAIRE antibodies. PSTAIRE served as a loading control. Relative intensity values normalized to PSTAIRE are indicated below each lane. (C) Logarithmically grown MSY001 cells expressing GFP-Mtw1 were fixed and stained with DAPI (blue) and anti-GFP antibodies (pseudo-colored in red). Scale bar, 2.5 µm. (D) Cell cycle stage-specific localization dynamics of GFP-Mtw1. Scale bar, 2.5 µm.

The online version of this article includes the following source data and figure supplement(s) for figure 2:

**Figure supplement 1.** Identification of kinetochore proteins in *M. sympodialis* by BLAST.

**Figure supplement 1—source data 1.** Identification of kinetochore proteins in *M. sympodialis* by BLAST.

## Mtw1 is localized to a single region at the GC minima of each *M. sympodialis* chromosome

Having identified Mtw1 as an authentic kinetochore protein, we performed ChIP-sequencing using the GFP-Mtw1 expressing strain of *M. sympodialis* (MSY001). Mapping the reads to the reference genome of *M. sympodialis* strain ATCC42132 (*Zhu et al., 2017*) identified one significantly enriched locus on each of the eight chromosomes (*Figure 3A*). The lengths of the Mtw1-enriched centromere regions identified from the ChIP-seq analysis ranged from 3167 bp to 5143 bp with an average length of 4165 bp (*Table 1*). However, the region of maximum Mtw1 enrichment on each

**Table 1.** Coordinates of centromeres and their GC content in *M. sympodialis.*
Coordinates and length of Mtw1-enriched regions in comparison with those of the core centromeres in *M. sympodialis.*

| Chromosome number | Core centromere | | | | Full-length centromere | | |
| | Coordinates | | | | Coordinates | | |
| | Start | End | Length (bp) | %GC | Start | End | Length (bp) |
|---|---|---|---|---|---|---|---|
| 1 | 786,541 | 787,061 | 520 | 16.4 | 784,833 | 788,599 | 3767 |
| 2 | 355,760 | 355,841 | 81 | 20 | 354,218 | 357,486 | 3269 |
| 3 | 237,534 | 238,686 | 1152 | 15.6 | 235,615 | 239,940 | 4326 |
| 4 | 418,202 | 418,728 | 526 | 15.2 | 415,985 | 420,656 | 4672 |
| 5 | 125,056 | 125,220 | 164 | 18 | 123,219 | 127,284 | 4066 |
| 6 | 101,950 | 102,502 | 552 | 14.4 | 100,342 | 105,251 | 4910 |
| 7 | 431,542 | 431,987 | 445 | 13.2 | 430,028 | 433,194 | 3167 |
| 8 | 24,694 | 25,564 | 870 | 18.4 | 22,334 | 27,476 | 5143 |

Genome average GC content (in %): 58.5.

chromosome (based on the number of sequenced reads aligned) mapped to the intergenic region harboring the GC trough (approximately 1 kb long), which was previously predicted to be the centromeres of *M. sympodialis* (*Figure 3B*) (*Zhu et al., 2017*). The regions of Mtw1 enrichment span beyond the core centromeres and include active genes located proximal to these troughs (*Figure 3B*, *Figure 3—figure supplement 1A*). However, these open reading frames (ORFs) do not show consensus features such as the orientation of transcription or functional classification. We validated this enrichment by ChIP-qPCR analysis with primers homologous to the predicted centromeres compared to those homologous to a control region distant from the centromere (*Figure 3—figure supplement 1B*).

## Histone H3 is depleted at the core centromere with active genes at the pericentric regions in *M. sympodialis*

The presence of CENP-A nucleosomes at the kinetochore should result in reduced histone H3 enrichment at the centromeres when compared to a non-centromeric locus. To test this, we performed ChIP with anti-histone H3 antibodies and analyzed the immunoprecipitated (IP) DNA by qPCR. As compared to a control ORF region that is unlinked to the centromere (190 kb away from *CEN1*), the pericentric regions flanking the core centromere showed a marginal reduction in histone H3 enrichment, which was further reduced at the core, that maps to the GC trough with the highest enrichment of the kinetochore protein Mtw1. That the core centromere region showing the maximum depletion of histone H3 coincided with the regions most enriched with Mtw1 further supports that histone H3, in these regions, is possibly replaced by its centromere-specific variant CENP-A (*Figure 3C*).

## The short regional centromeres of *M. sympodialis* are enriched with a 12-bp-long AT-rich consensus sequence motif

To understand the features of *M. sympodialis* centromeres, we analyzed the centromere DNA sequences for the presence of consensus motifs or structures such as inverted repeats. PhyloGibbs-MP (*Siddharthan et al., 2005*; *Siddharthan, 2008*) predicted a 12-bp-long AT-rich motif common to all of the centromere sequences of *M. sympodialis* (*Figure 3D*). We swept the PWM from the Phylo-Gibbs-MP output across each chromosome of *M. sympodialis* and counted the number of motif predictions in a sliding 500-bp window, sliding by 100 bp at a time. Sites with log-likelihood-ratio (LLR) of >7.5 were counted as motif predictions. The LLR is the natural logarithm of the ratio of the likelihood of the 12-bp substring arising as a sample from the PWM to the likelihood of it being generic 'background'. In each case, the global peak coincides with the centromere (*Figure 3—figure supplement 2A*). In each chromosome, the centromere region shows between 7 and 13 motif matches, whereas no other 500-bp window shows more than three matches. This suggests that the AT-rich motif is more enriched at the centromeres than at any other region in the *M. sympodialis* genome

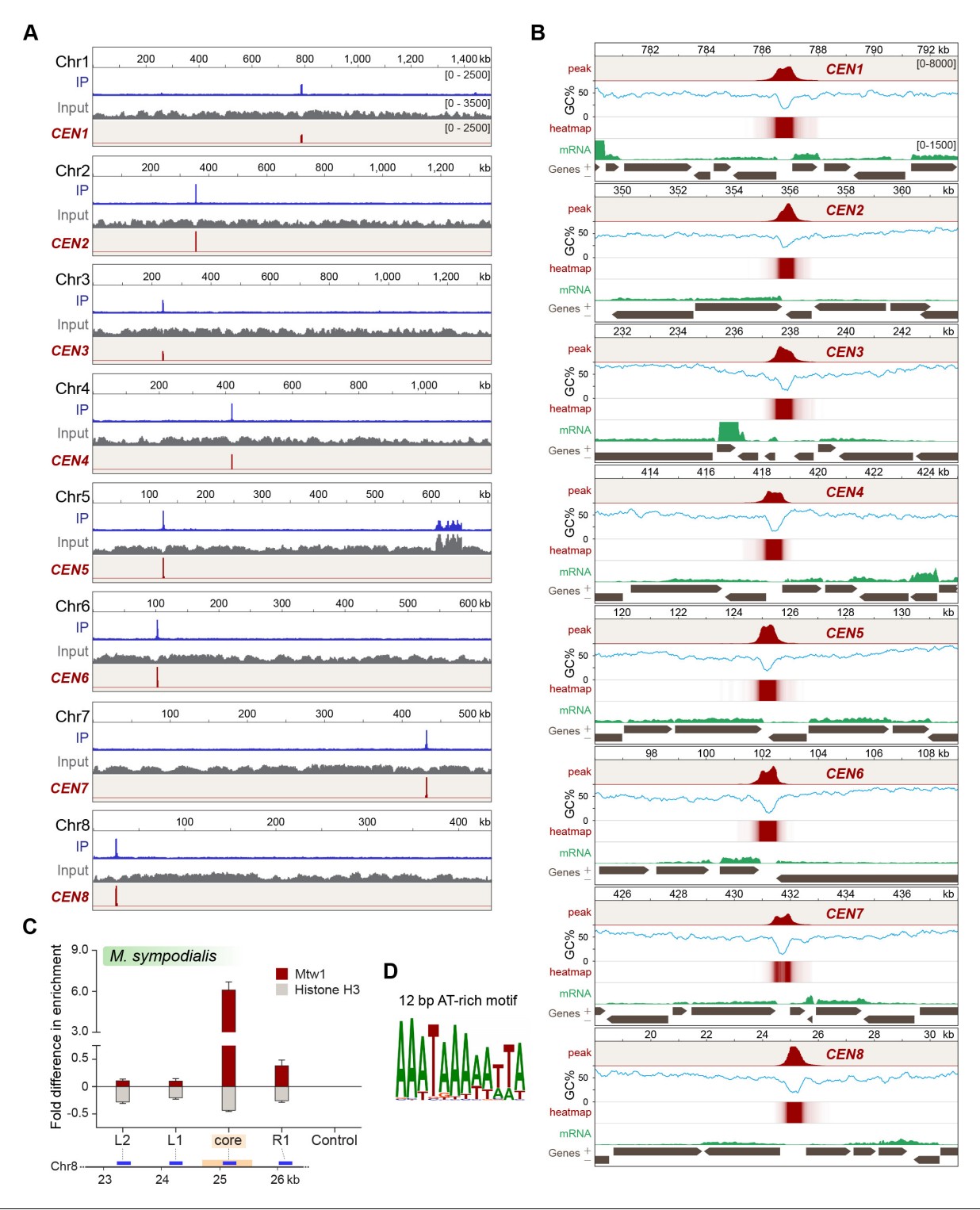

**Figure 3.** Localization of Mtw1 to single-peaks identifies centromeres on each of the eight chromosomes of *M. sympodialis*. (**A**) GFP-Mtw1 ChIP-seq reads were mapped to each chromosome. The *x*-axis indicates chromosomal coordinates (in kb) and the *y*-axis indicates read depth. 'Input', reads from total DNA; 'IP,' reads from immunoprecipitated sample; *CEN,* Mtw1-enriched regions derived by subtracting input reads from those of the IP sample (peak values 0–2500). Additional peaks observed in both IP and input tracks on Chr5 are from the rDNA locus. (**B**) A 13 kb-window of the Mtw1 enrichment profile (*CEN*, represented as peaks and heat-map in two different tracks, red) plotted along with the GC content (%GC, blue) and regions of transcription (RNA-seq, green). Numbers in the topmost track in every panel indicate chromosomal coordinates (in kb). The scales for the *y*-axis are as follows: *CEN* (0–8000), %GC (0–75), RNA-seq reads (0–1500). Gray arrows in each panel indicate predicted ORFs based on RNA-seq data with

*Figure 3 continued on next page*

*Figure 3 continued*

arrowheads indicating the direction of transcription of the corresponding gene, also marked as '+' and '–' in the axis label. (C) Fold difference in Mtw1 and histone H3 enrichment at *CEN8*, as compared to a non-centromeric control region (190 kb away on the right of *CEN1*) by qPCR analysis. Schematic of a 4-kb region of Chr8 with the *CEN8* core (yellow box) is depicted below the graph. Blue lines indicate regions assayed by PCR: core-region corresponding to the GC trough; L1 and R1, 750 bp away from the core; L2, 1500 bp away from the core; and a non-centromeric control region (190 kb away from centromere in Chr1). The *x*-axis indicates regions across the *CEN8* probed by PCR and the *y*-axis indicates fold difference in the enrichment of Mtw1 and histone H3 as compared to the control region. Error bars indicate standard deviations (SD). Values from three experiments, each performed with three technical replicates, were used to generate the plot. (D) Logo of the consensus DNA sequence identified from *M. sympodialis* centromeres, graphically represented with the size of the base corresponding with the frequency of occurrence.

The online version of this article includes the following source data and figure supplement(s) for figure 3:

**Source data 1.** Source raw data for *Figure 3C* (ChIP-qPCR for GFP-Mtw1 and Histone H3 across *MsyCEN8*).
**Figure supplement 1.** Mtw1-enriched regions in *M. sympodialis* contain transcriptionally active genes.
**Figure supplement 1—source data 1.** Source raw data for *Figure 3—figure supplement 1B* (ChIP-qPCR for GFP-Mtw1 across all *M. sympodialis* centromeres).
**Figure supplement 2.** Sequence features of centromeres in *M. sympodialis*.
**Figure supplement 3.** Enrichment of the 12-bp motif at the centromere core in *M. sympodialis*.

---

(*Figure 3—figure supplement 2A*). To ensure that this is not an artifact of the GC-poor nature of the centromere, we repeated the analysis with a synthetic shuffled PWM, created by scrambling the order of the columns of the original PWM (that is, scrambling the positions in the motif while keeping the corresponding weight vectors the same). This shuffled motif showed more matches in the centromeres than are seen in the non-centromeric genomic sequence, but significantly fewer than are seen in the authentic centromeric sequences of most chromosomes (*Figure 3—figure supplement 2B*). A dot-plot analysis was performed to detect the presence of any direct or inverted-repeat structure associated with the centromeres in *M. sympodialis*. Analysis of all of the centromere sequences and 5-kb flanking sequences using SyMap confirmed the lack of direct/inverted repeat structures (*Figure 3—figure supplement 2C*).

In the absence of any centromere-exclusive DNA sequence, the unique and distinguishing features of centromere regions in *M. sympodialis* are an AT-rich core region of <1 kb (average AT content of 78% as compared to the genome average of 41.5%) that is enriched with the 12-bp motif (*Figure 3—figure supplement 3*) within a kinetochore protein-bound region of 3–5 kb. As expected, the kinetochore-bound region contains a reduced level of histone H3.

## Centromeres in *M. furfur*, *M. slooffiae*, and *M. globosa* map to chromosomal GC minima

Using the unique centromere features identified in *M. sympodialis*, we predicted one centromere locus on each of the seven *M. furfur* chromosomes, and these all map to chromosomal GC troughs (*Figure 4—figure supplement 1A*, *Table 2*). We also predicted the centromeres in *M. slooffiae*, *M. globosa*, and *M. restricta*, each of which contains nine chromosomes (*Figure 4—figure supplement 1B–D*, *Table 2*). Each of the predicted centromere regions is enriched with the 12-bp AT-rich motif identified in *M. sympodialis* centromere sequences as compared to other regions in the genomes (*Figure 4—figure supplement 2*).

To validate the centromeric loci in *M. furfur* experimentally, we functionally expressed the centromeric histone H3 variant CENP-A with a 3xFLAG tag at the C-terminus (*Figure 4A*). We performed ChIP in strain MF001 (CENP-A-3xFLAG) and analyzed immunoprecipitated DNA by qPCR using primers specific to each of the seven predicted centromeres and to a centromere-unlinked control locus 1.3 Mb away from *CEN1*. Enrichment of CENP-A at all seven centromeres over the control locus confirmed that the predicted regions are indeed centromeres in *M. furfur* CBS14141 (*Figure 4B,C*).

Given the lack of genetic manipulation methods for *M. slooffiae* and *M. globosa*, we tested the enrichment of histone H3 at the predicted centromeres in these two species. All of the nine centromeric loci in these two species contained a reduced histone H3 level when compared to a control locus that was unlinked to centromeres (*Figure 4D,E*). Furthermore, upon analyzing the enrichment profile at one centromere (*CEN1*) in *M. slooffiae*, we observed a reduction in the enrichment levels of histone H3 at the GC troughs as compared to the flanking regions (*Figure 4F*). In the case of *M.*

**Table 2.** Coordinates, length, and GC content (in %) of the centromeres predicted in *M. furfur, M. globosa, M. slooffiae,* and *M. restricta.*

| | Chr./scaffold | CEN | Core centromere | | | | % GC genome |
| | | | Start | End | Length (bp) | % GC | |
|---|---|---|---|---|---|---|---|
| *M. furfur* CBS14141 | Chr1 | CEN1 | 2,850,135 | 2,850,402 | 268 | 15.7 | 64.9 |
| | Chr2 | CEN2 | 68,763 | 68,931 | 168 | 15.4 | |
| | Chr3 | CEN3 | 717,557 | 718,084 | 528 | 22.9 | |
| | Chr4 | CEN4 | 155,897 | 156,301 | 405 | 18.3 | |
| | Chr5 | CEN5 | 342,885 | 343,372 | 488 | 21.5 | |
| | Chr6 | CEN6 | 86,112 | 86,832 | 721 | 27 | |
| | Chr7 | CEN7 | 56,894 | 57,339 | 445 | 20.9 | |
| *M. globosa* CBS7966 | Chr1 | CEN1 | 981,894 | 982,242 | 349 | 17.7 | 52.05 |
| | Chr2 | CEN2 | 362,480 | 362,807 | 327 | 25.9 | |
| | Chr3 | CEN3 | 219,647 | 220,121 | 474 | 27.2 | |
| | Chr4 | CEN4 | 152,635 | 152,994 | 359 | 18.3 | |
| | Chr5 | CEN5 | 215,437 | 215,595 | 158 | 17 | |
| | Chr6 | CEN6 | 464,007 | 464,114 | 107 | 32.4 | |
| | Chr7 | CEN7 | 736,701 | 737,015 | 314 | 18.1 | |
| | Chr8 | CEN8 | 59,472 | 59,817 | 345 | 19.7 | |
| | Chr9 | CEN9 | 114,080 | 114,535 | 455 | 23.5 | |
| *M. slooffiae* CBS7956 | Chr1 | CEN1 | 138,919 | 139,465 | 547 | 26 | 66.31 |
| | Chr2 | CEN2 | 132,717 | 133,193 | 477 | 23.1 | |
| | Chr3 | CEN3 | 367,665 | 368,177 | 513 | 23.8 | |
| | Chr4 | CEN4 | 130,942 | 131,501 | 560 | 27 | |
| | Chr5 | CEN5 | 183,442 | 183,981 | 540 | 28.5 | |
| | Chr6 | CEN6 | 411,984 | 412,552 | 569 | 27.4 | |
| | Chr7 | CEN7 | 54,307 | 54,889 | 583 | 30 | |
| | Chr8 | CEN8 | 497,637 | 498,149 | 513 | 24 | |
| | Chr9 | CEN9 | 55,948 | 56,479 | 532 | 26.7 | |
| *M. restricta* CBS877 | Chr1 | CEN1 | 347,813 | 348,406 | 594 | 29.7 | 55.73 |
| | Chr2 | CEN2 | 87,190 | 87,806 | 617 | 33.3 | |
| | Chr3 | CEN3 | 1,101,494 | 1,102,083 | 590 | 33.9 | |
| | Chr4 | CEN4 | 754,356 | 754,989 | 634 | 34.2 | |
| | Chr5 | CEN6 | 621,177 | 621,863 | 687 | 31.7 | |
| | Chr6 | CEN7 | 390,657 | 391,286 | 630 | 35.1 | |
| | Chr7 | CEN8 | 362,842 | 363,381 | 540 | 32 | |
| | Chr8 | CEN9 | 117,021 | 117,603 | 583 | 32.8 | |
| | Chr9 | CENR | 70,306 | 70,913 | 608 | 36.3 | |

*globosa*, the regions spanning a centromere (*CEN2*) also depicted a similar reduction in the histone H3 levels (*Figure 4G*). Taken together, the significant reduction in the histone H3 levels at the predicted centromeres, indicative of the presence of CENP-A, suggests that these putative centromere regions are indeed bona fide centromeres in these species.

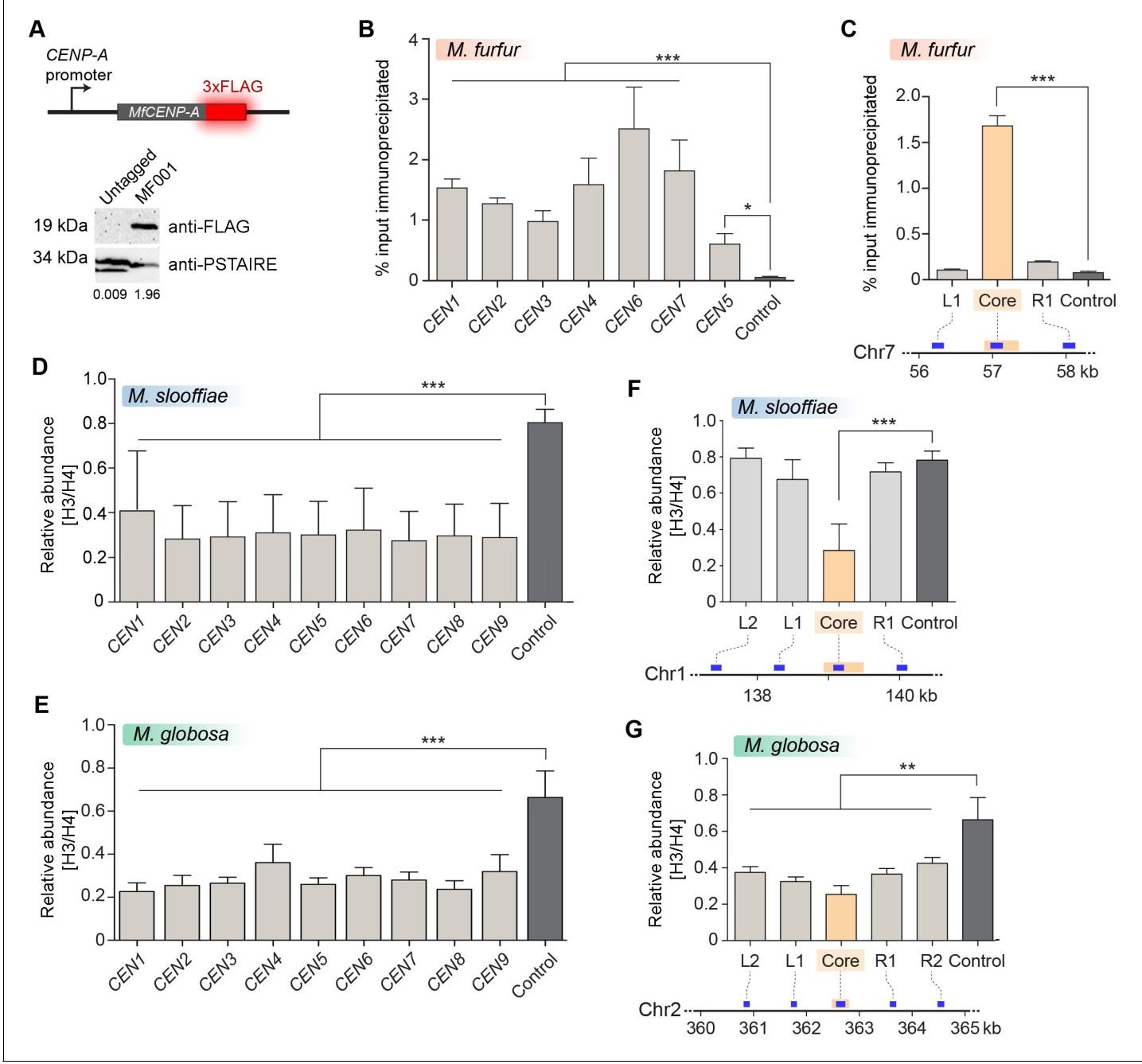

**Figure 4.** Centromeres in *M. furfur, M. slooffiae,* and *M. globosa* map to global GC troughs on each chromosome. (**A**) Schematic of epitope tagging of CENP-A with 3xFLAG at the C-terminus. (Bottom) Immunoblot analysis using whole-cell lysates prepared from the untagged wild-type *M. furfur* (CBS14141) cells and CENP-A-3xFLAG expressing cells (MF001) probed with anti-FLAG antibodies and anti-PSTAIRE antibodies. PSTAIRE served as the loading control. (**B**) Abundance of CENP-A at each of the predicted *M. furfur* centromeres by qPCR analysis of DNA immunoprecipitated with anti-FLAG affinity gel in MF001 cells expressing CENP-A-3xFLAG. The *x*-axis indicates individual *CEN* regions assayed with primers that are homologous to the GC troughs on each chromosome that were predicted to be centromeres (see *Supplementary file 1* for primer sequences). The non-centromeric control sequence maps to a region 1.3 Mb away from predicted *CEN1*. The *y*-axis indicates enrichment of CENP-A estimated as the percentage of input immunoprecipitated. (**C**) Abundance of CENP-A across *MfCEN7* by ChIP-qPCR analysis in MF001 cells. A schematic representation of a 2-kb region is shown below the graph. The yellow bar indicates the centromere core of *CEN7* corresponding to the GC trough. Blue bars indicate regions analyzed by qPCR: L1 and R1, which are 750 bp away from the centromere core. 'Control' refers to a region 1.3 Mb away from *CEN1*. CENP-A enrichment is plotted along the *y*-axis as the percentage of input immunoprecipitated. (**D, E**) Comparison of the relative abundance of histone H3 compared to histone H4 at the predicted centromeres to a non-centromeric control locus in *M. slooffiae* (D) and *M. globosa* (E), respectively. Enrichment was estimated as the percentage of input immunoprecipitated with histone H3 and histone H4 antibodies and their ratio is plotted as

*Figure 4 continued on next page*

*Figure 4 continued*

relative enrichment (*y*-axis). The *x*-axis indicates centromeres in each species and these were assayed with primers homologous to GC troughs (i. e. predicted centromeres) in each chromosome (see *Supplementary file 1* for primer sequences). The control region that is unlinked to the centromere corresponds to a locus 630 kb away from *CEN1* in *M. slooffiae* and 416 kb away from *CEN2* in *M. globosa*. (F) The relative abundance of histone H3 compared to histone H4 across *MslCEN1* as determined by qPCR analysis of the DNA immunoprecipitated using histone H3 and histone H4 antibodies. A schematic of the *MslCEN1* locus is shown below the graph. The yellow bar indicates the *CEN1* core region corresponding to the GC trough. Blue bars indicate regions analyzed by qPCR: L1 and L2 map to regions 750 bp and 1.5 kb to the left of the *CEN1* core; R1 maps to a region 750 bp to the right of the *CEN1* core. The control region corresponds to a locus 630 kb away from the *CEN1* core. The ratio of enrichment of histone H3 to that of histone H4 is plotted as the relative enrichment (*y*-axis). (G) The relative abundance of histone H3 compared to histone H4 across *MgCEN2* by qPCR analysis of the DNA immunoprecipitated using histone H3 and histone H4 antibodies. A schematic of a 5-kb region containing *MgCEN2* is shown below the graph. The yellow bar indicates the *CEN2* core region corresponding to the GC trough. Blue bars indicate regions analyzed by qPCR: L1 and L2 indicate regions 750 bp and 1.5 kb to the left of the *CEN2* core; R1 and R2 indicate regions 750 bp and 1.5 kb to the right of the *CEN2* core. The control region corresponds to a locus 416 kb away from the *CEN2* core. The ratio of enrichment of histone H3 to that of histone H4 is plotted as the relative enrichment (*y*-axis). Values from three experiments, each performed with three technical replicates, were used to generate the plots in panels (B–G). Error bars indicate standard deviations (SD). Statistical significance was tested by one-way ANOVA: *, significant at $p < 0.05$, ***, significant at $p < 0.001$.

The online version of this article includes the following source data and figure supplement(s) for figure 4:

**Source data 1.** Source raw data for *Figure 4B* (ChIP-qPCR for CENP-A-3xFLAG across all *M. furfur* centromeres).
**Source data 2.** Source raw data for *Figure 4C* (ChIP-qPCR for CENP-A and Histone H3 across *MfCEN7*).
**Source data 3.** Source raw data for *Figure 4D* (ChIP-qPCR for Histone H3 and Histone H4 across all *M. slooffiae* centromeres).
**Source data 4.** Source raw data for *Figure 4E* (ChIP-qPCR for Histone H3 and Histone H4 across all *M. globosa* centromeres).
**Source data 5.** Source raw data for *Figure 4F* (ChIP-qPCR for Histone H3 and Histone H4 across *MslCEN1*).
**Source data 6.** Source raw data for *Figure 4G* (ChIP-qPCR for Histone H3 and Histone H4 across *MgCEN2*).
**Figure supplement 1.** Putative centromeres of *M. furfur, M. slooffiae, M. globosa,* and *M. restricta* map to global GC troughs in each chromosome.
**Figure supplement 2.** The 12-bp AT-rich motif is enriched at the putative centromeres of *M. furfur, M. slooffiae, M. globosa,* and *M. restricta*.

## Centromere loss by breakage resulted in chromosome number reduction in *M. sympodialis*

Synteny of genes across centromeres is largely conserved in closely related species (*Byrne and Wolfe, 2005*; *Padmanabhan et al., 2008*; *Yadav et al., 2018b*). To understand the transition between different chromosome number states, we analyzed the conservation of gene synteny across centromeres in these species. By mapping gene synteny at the centromeres of *M. globosa* and *M. slooffiae* (each with nine chromosomes), compared with that of *M. sympodialis* (containing eight chromosomes), we found complete gene synteny conservation in eight of the nine centromeres (*Figure 5A,B*). Thus, syntenic regions of all eight *M. sympodialis* centromeres are present in the genomes of *M. globosa* and *M. slooffiae*. In the case of *M. restricta*, seven putative centromeres are completely syntenic with *M. sympodialis* centromeres and one centromere retained partial gene synteny (*Table 3*). However, no gene synteny conservation was observed at the centromeres of Chr2 in *M. globosa*, Chr5 in *M. slooffiae*, or Chr8 in *M. restricta* (*Table 3*), indicating the loss of a centromere during the transition from the nine-chromosome state to the eight-chromosome state.

The GC trough corresponding to *MgCEN2/MslCEN5* is flanked by genes that map to MsyChr2 on one arm and MsyChr4 on the other (*Figure 5C,D*). The centromere region in each of MgChr2 and MslChr5 marks a synteny breakpoint showing no homologous region in the *M. sympodialis* genome, indicating a loss of this centromere DNA sequence. We also observed that the genes flanking the breakpoint are conserved in *M. sympodialis*, suggesting that the specific intergenic region was involved (*Figure 5E,F*). Evidence for the internalization of telomere-adjacent ORFs or for the presence of interstitial telomere repeats that are indicative of telomere–telomere fusions was not detected in the *M. sympodialis* genome. These observations strongly support our hypothesis that breakage of *MgCEN2/MslCEN5* (or the orthologous ancestral *CEN*) and fusion of the two acentric arms to other chromosomes resulted in the reduction in chromosome number observed when comparing these species.

## Centromere inactivation by sequence divergence and loss of AT-richness resulted in chromosome number reduction in *M. furfur*

To understand the basis of the change in chromosome number from nine to seven in *Malassezia* species, we compared the synteny of ORFs flanking the *M. slooffiae* or *M. globosa* centromeres with

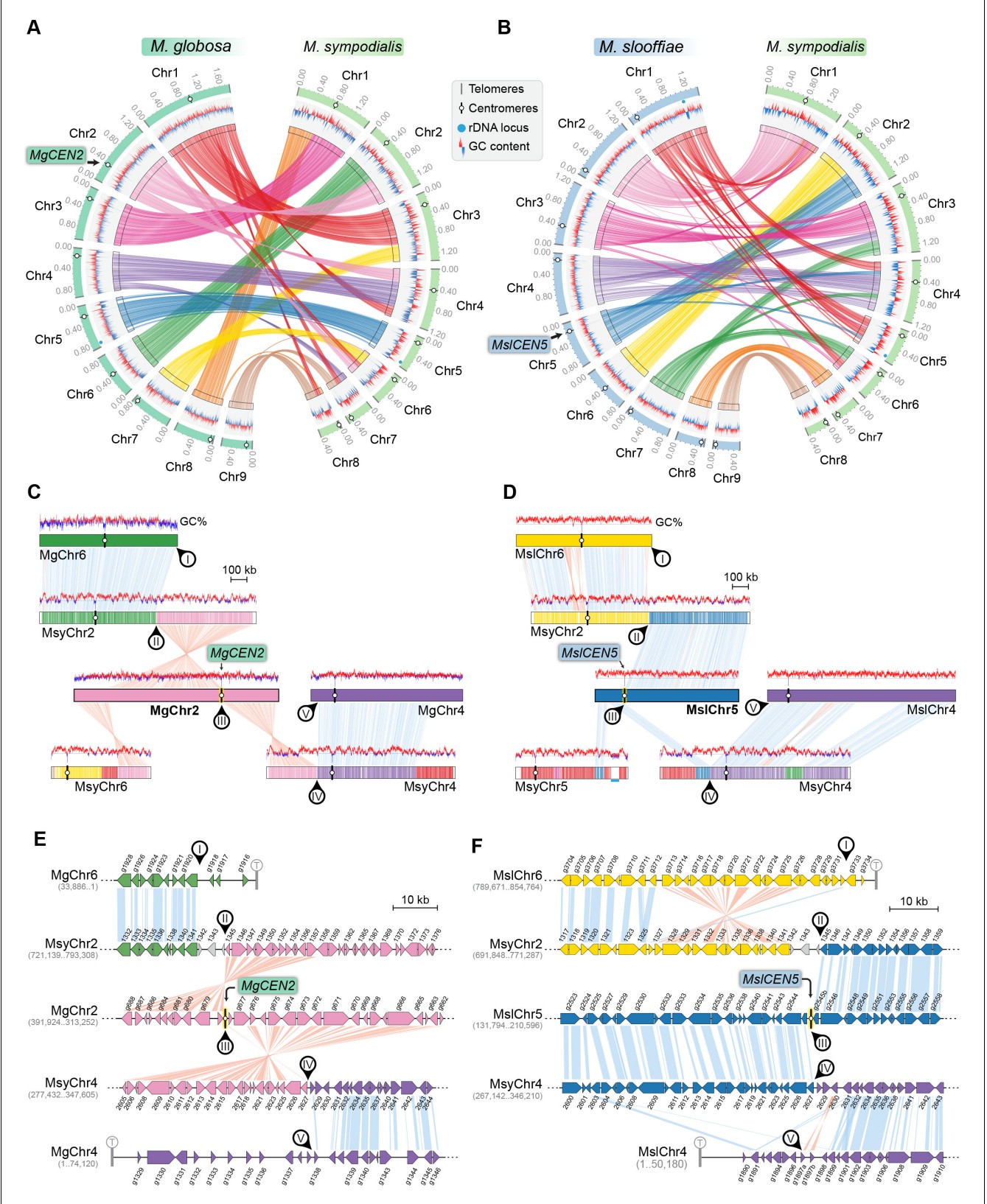

**Figure 5.** *MgCEN2* and *MslCEN5* map to a synteny breakpoint in *M. sympodialis*. (**A, B**) Circos plots depicting the gene synteny blocks that are conserved between *M. globosa* and *M. sympodialis* chromosomes and between *M. slooffiae* and *M. sympodialis* chromosomes. Tracks from outside to inside represent the positions of centromeres and telomeres, GC content (plotted as blue and red lines indicating GC content below or above genome

*Figure 5 continued on next page*

*Figure 5 continued*

average, calculated in 0.4-kb non-overlapping windows), and colored connectors indicate regions of conserved gene synteny between the two species. (C, D) Linear chromosome plots depicting syntenic regions between Chr2 of *M. globosa* and Chr5 of *M. slooffiae* with chromosomes of *M. sympodialis*. GC content (in %) is shown as red/blue lines above each chromosome. Circular labels mark the gene synteny breakpoints. The synteny breakpoint at *MgCEN2* and *MslCEN5* is marked as III. The regions on MsyChr2 and MsyChr4 where the homologs of ORFs flanking the breakpoint are located are marked II and IV, respectively. Labels I and V indicate gene synteny conservation on the other side of the fusion points II and IV on MsyChr2 and MsyChr4 as compared to *M. globosa* and *M. slooffiae* chromosomes, respectively. (E, F) Zoomed-in image of the gene synteny breakpoint at *MgCEN2* and *MslCEN5*, representing the conservation of genes flanking these centromeres in *M. sympodialis* chromosomes at the ORF level.

that of *M. furfur*. Of the nine centromeres in *M. slooffiae*, three centromeres belonged to conserved gene synteny blocks and four others retained partial gene synteny conservation in *M. furfur* (*Figure 6A*, *Table 3*). A similar pattern of gene synteny conservation was observed between *M. globosa* and *M. furfur* (*Figure 6B*, *Table 3*). The genes flanking the remaining two centromeres (*CEN8* and *CEN9*) in *M. slooffiae* were present in conserved gene synteny blocks in the two arms of MfChr3 (*Figure 6C*). However, the regions corresponding to *CEN8* and *CEN9* in *M. slooffiae* appear to have evolved to decreased AT-richness in *M. furfur*. A similar centromere inactivation mechanism was observed when *CEN8* and *CEN9* of *M. globosa* were compared to the corresponding syntenic regions in *M. furfur* (*Figure 6D*). These results suggest centromere inactivation by changes in the centromeric DNA sequence in this species (*Figure 6—figure supplement 1*).

## The common ancestral *Malassezia* species contained nine chromosomes

To trace the ancestral karyotype in *Malassezia*, we predicted centromeres and inferred chromosome numbers in other species of clades A and B on the basis of GC troughs and gene synteny. We identified putative centromeres in *M. dermatis* and *M. nana* in Clade B because of their relatively better-assembled genomes distributed in 18 and 13 scaffolds, respectively. Of these, we could predict eight centromeric regions marked by GC troughs that were also enriched with the 12-bp motif in each species (*Figure 7—figure supplement 1A–B*, *Figure 7—figure supplement 2A–B* and *Table 4*). Furthermore, in both of these species, the eight putative centromeres shared complete gene synteny with the regions spanning *M. sympodialis* centromeres, indicating that their common ancestor had eight chromosomes (green circle in *Figure 7A*, *Figure 7—figure supplement 3* and *Table 3*). To map the common ancestor in Clade B *Malassezia* species, we analyzed regions flanking centromeres of Chr2 of *M. globosa* and Chr8 of *M. restricta*, both of which mapped to the gene synteny breakpoint of the genome of the *Malassezia* species that have eight chromosomes suggesting that their common ancestor, named as Ancestor B (Anc. B), also had nine chromosomes (*Figure 7A*, *Figure 7—figure supplement 3*). On the basis of our centromere predictions in Clade B species and synteny analysis, we propose that centromere breakage would have occurred in the

**Table 3.** Synteny of centromeres across all of the *Malassezia* species analyzed in this study.

| Clade C | Clade B1 | | Clade B2 | | | | Clade A | |
|---|---|---|---|---|---|---|---|---|
| *M. slooffiae* (9 Chr) | *M. dermatis* (8 Chr) | *M. nana* (8 Chr) | *M. sympodialis* (8 Chr) | *M. globosa* (9 Chr) | *M. restricta* (9 Chr) | *M. vespertilionis* (9 Chr) | *M. japonica* (9 Chr) | *M. furfur* (7 Chr) |
| *CEN2* | *CEN1* | *CEN1* | *CEN1* | *CEN3* | *CEN5* partial | *CEN6* | *CEN8* | *CEN4* |
| *CEN6* | *CEN2* | *CEN2* | *CEN2* | *CEN6* | *CEN7* | *CEN9* | *CEN4* partial | *CEN2* partial |
| *CEN3* | *CEN3* | *CEN3* | *CEN3* | *CEN1* | *CEN3* | *CEN7* partial | *CEN6* partial | *CEN3* partial |
| *CEN4* | *CEN4* | *CEN4* | *CEN4* | *CEN4* | *CEN1* | *CEN1* | *CEN2* | *CEN7* partial |
| *CEN1* | *CEN6* | *CEN6* | *CEN5* | *CEN5* | *CEN2* | *CEN4* | *CEN7* | *CEN1* |
| *CEN7* | *CEN5* | *CEN5* | *CEN6* | *CEN7* | *CEN6* | *CEN5* partial | *CEN5* partial | *CEN6* |
| *CEN8* | *CEN7* | *CEN7* | *CEN7* | *CEN9* | *CEN4* | *CEN2* | *CEN9* partial | Inactivated |
| *CEN9* | *CEN8* | *CEN8* | *CEN8* | *CEN8* | *CEN9* | *CEN8* | *CEN1* | Inactivated |
| *CEN5* | BP | BP | BP | *CEN2* | *CEN8* partial | *CEN3* | *CEN3* partial | *CEN5* |

'BP' indicates the presence of a gene synteny break. 'Inactivated' indicates centromere inactivation resulting from sequence divergence and erosion of AT-richness.

common ancestor of *M. sympodialis*, *M. nana,* and *M. dermatis* after divergence from the common ancestor of *M. globosa* and *M. restricta*, which retained a nine-chromosome configuration (*Figure 7A,B*).

As mentioned earlier, *M. furfur* and *M. obtusa* of Clade A contain seven chromosomes each (*Boekhout and Bosboom, 1994*; *Boekhout et al., 1998*). To further understand the karyotype variations within this clade, we predicted the chromosome number in *Malassezia vespertilionis* and *Malassezia japonica* because their genomes are relatively well assembled (*Sugita et al., 2003*; *Lorch et al., 2018*). In both of these species, we were able to predict nine GC troughs, indicative of the centromeres of nine chromosomes (*Figure 7—figure supplement 1C–D*, *Table 4*). In the case of *M. vespertilionis*, all of the predicted centromeres showed enrichment of the 12-bp motif (*Figure 7— figure supplement 2C*). However, the 12-bp motif was found to be enriched in all of the predicted centromeres except the centromere of scaffold 7 of *M. japonica* (*Figure 7—figure supplement 2D*). The presence of species with nine chromosomes in Clade A suggests that the ancestral state in this clade, Anc. A, also contained nine chromosomes (*Figure 7A*).

We identified nine centromeres in *M. slooffiae*, the only species in Clade C with a well-assembled genome. The presence of species with nine chromosomes in each of the three clades of *Malassezia* species, the conservation of gene synteny across orthologous centromeres, and the similar centromere features shared by all nine species analyzed in this study suggest that *Malassezia* species diverged from a common ancestor that had nine chromosomes with short regional centromeres enriched with the 12-bp AT-rich DNA sequence motif.

## Discussion

In this study, we experimentally validated the chromosome number in *M. slooffiae* and *M. globosa* by PFGE analysis. We sequenced and assembled the genomes of *M. slooffiae*, *M. globosa*, and *M. furfur* and compared each one with the genome of *M. sympodialis* in order to understand the karyotype differences observed in members of the *Malassezia* species complex. These species represent each of the three major clades of *Malassezia* species with chromosome numbers ranging from seven to nine. Because centromere loss or gain directly influences the chromosome number of a given species, we experimentally identified the centromeres of these representative species to understand the mechanisms of karyotype diversity. Kinetochore proteins are useful tools in identifying the centromeres of an organism. The localization of the evolutionarily conserved kinetochore protein Mtw1 suggested that kinetochores are clustered throughout the cell cycle in *M. sympodialis*. ChIP-sequencing analysis identified short regional (<5 kb long) centromeres in *M. sympodialis* that are depleted of histone H3, but enriched with an AT-rich sequence motif. The identification of centromeres in *M. slooffiae*, *M. globosa*, and *M. furfur* further suggested that centromere properties are conserved across these *Malassezia* species. By predicting putative centromeres in five other species and by considering four species with experimentally mapped centromeres described above across three clades of *Malassezia*, we concluded that an AT-rich centromere core of <1 kb in length enriched with the 12-bp sequence motif is a potential signature of centromeres in the nine *Malassezia* species analyzed in this study. Comparative genomics analysis revealed two major mechanisms of centromere inactivation resulting in karyotype change. The presence of a nine-chromosome state in each of the three clades, along with conserved centromere features and conserved gene synteny, helped us infer that the ancestral *Malassezia* species had nine chromosomes that had short regional centromeres with an AT-rich core enriched with the 12-bp sequence motif.

Centromeres in the *Malassezia* species complex represent the first example of short regional centromeres in basidiomycetes. Centromeres reported in other basidiomycetes, such as those in *Cryptococcus* and *Ustilago* species, are of the large regional type (*Yadav et al., 2018b*). The *Malassezia* species analyzed in this study have a significantly smaller genome (<9 Mb) than other basidiomycetes and lack RNAi machinery. The occurrence of short regional centromeres in RNAi-deficient *Malassezia* species is in line with a previous finding showing a reduction in centromere size in RNAi-deficient basidiomycete species as compared to their RNAi-proficient relatives (*Yadav et al., 2018b*). With the presence of clustered kinetochores across cell cycle stages, and the absence of key genes encoding the RNAi machinery, these *Malassezia* species resemble ascomycetes such as many of the CTG clade species with short regional centromeres (*Sanyal et al., 2004*; *Nakayashiki et al., 2006*; *Padmanabhan et al., 2008*; *Kapoor et al., 2015*; *Chatterjee et al., 2016*;

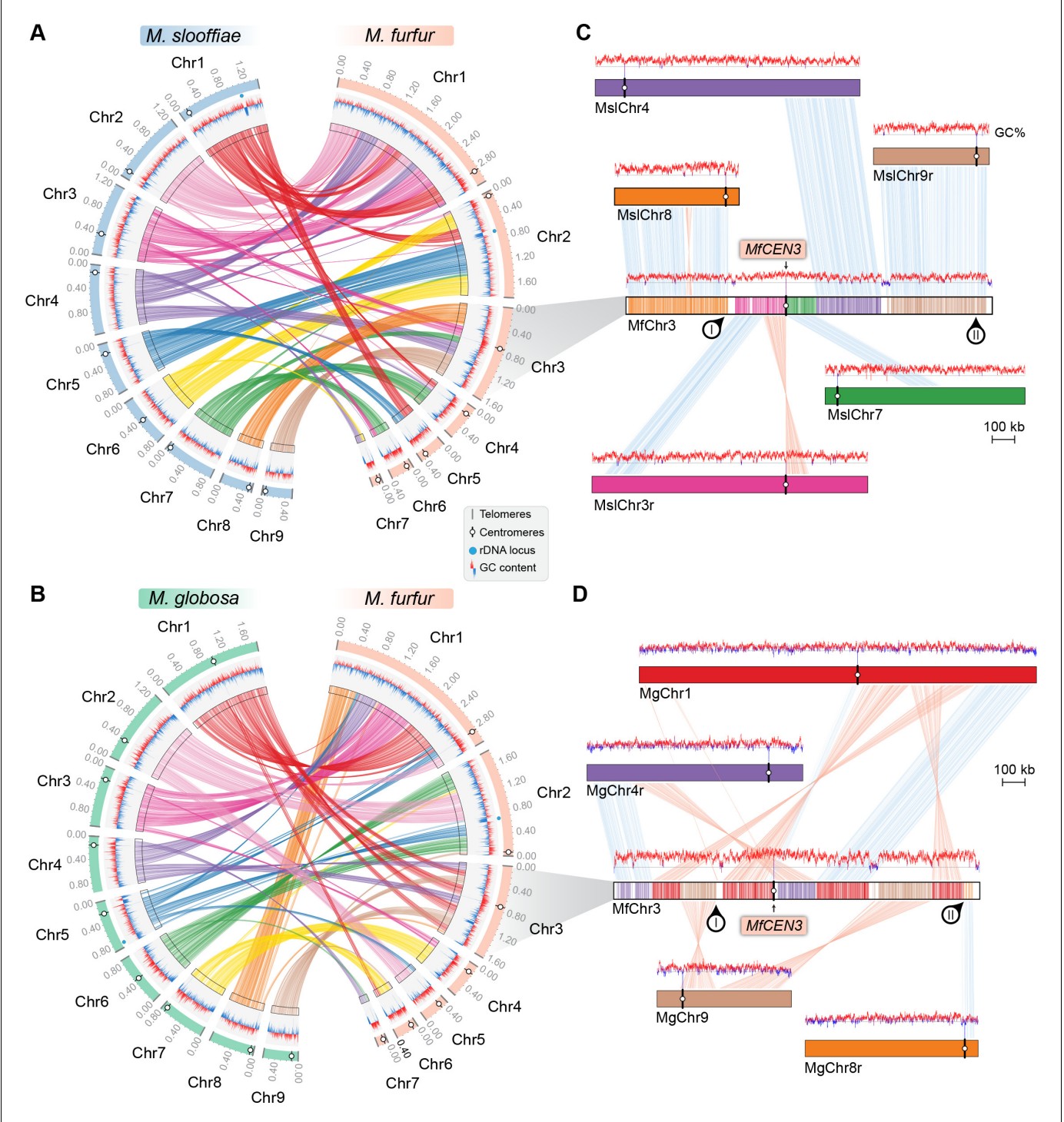

**Figure 6.** Inactivation of *CEN8* and *CEN9* of *M. slooffiae* and *M. globosa* in MfChr3 resulted in reduction in chromosome number in *M. furfur*. (**A, B**) Circos plots depicting the gene synteny blocks that are conserved between the *M. slooffiae* or *M. globosa* chromosomes and the *M. furfur* chromosomes. Tracks from outside to inside represent positions of centromeres and telomeres, GC content (plotted as blue and red lines indicating GC content below or above genome average, calculated in 0.4-kb non-overlapping windows), and colored connectors indicate regions of conserved gene synteny between each pair of two species. (**C**) Linear chromosome plot depicting the regions that show synteny between Chr3 of *M. furfur* and Chr3, Chr4, Chr7, Chr8, and Chr9 of *M. slooffiae*. GC content (in %) is shown as red/blue lines above each chromosome. Regions corresponding to *MslCEN8* and *MslCEN9* in MfChr3 are marked I and II, respectively. (**D**) Linear chromosome plot depicting the conservation of gene synteny between Chr3 of *M. furfur* and Chr1, Chr4, Chr8, and Chr9 of *M. globosa*. Regions corresponding to *MgCEN9* and *MgCEN8* in MfChr3 are marked I and II, respectively.

*Figure 6 continued on next page*

*Figure 6 continued*

The online version of this article includes the following figure supplement(s) for figure 6:

**Figure supplement 1.** Loss of AT-content in regions corresponding to *CEN8* and *CEN9* of *M. slooffiae* and *M. globosa* in MfChr3.

*Yadav et al., 2018a*), rather than the basidiomycetes with large regional centromeres. By combining these features, we conclude that the genome size and the presence of complete RNAi machinery could determine the centromere type of a species, irrespective of the phylum to which it belongs.

**Table 4.** Coordinates, length, and GC content (in %) of the centromeres predicted in *M. dermatis*, *M. nana*, *M. vespertilionis*, and *M. japonica*.

| | Chr./scaffold | CEN | Core centromere | | | | % GC genome |
| | | | Start | End | Length (bp) | % GC | |
|---|---|---|---|---|---|---|---|
| *M. dermatis* JCM11348 | BCKX01000001.1 (Scf1) | CEN1 | 711,456 | 711,978 | 523 | 22.8 | 59.05 |
| | BCKX01000002.1 (Scf2) | CEN2 | 1,014,281 | 1,014,977 | 697 | 31.7 | |
| | BCKX01000003.1 (Scf3) | CEN3 | 232,065 | 232,795 | 731 | 29.3 | |
| | BCKX01000004.1 (Scf4) | CEN4 | 409,839 | 410,631 | 793 | 29.5 | |
| | BCKX01000005.1 (Scf5) | CEN5 | 94,520 | 95,018 | 499 | 18.2 | |
| | BCKX01000006.1 (Scf6) | CEN6 | 473,487 | 474,334 | 848 | 30.4 | |
| | BCKX01000007.1 (Scf7) | CEN7 | 76,361 | 76,975 | 615 | 26 | |
| | BCKX01000008.1 (Scf8) | CEN8 | 17,893 | 18,540 | 648 | 26.4 | |
| *M. nana* JCM12085 | BCLA01000001.1 (Scf1) | CEN1 | 715,036 | 715,592 | 557 | 27.8 | 57.95 |
| | BCLA01000002.1 (Scf2) | CEN2 | 349,428 | 350,120 | 693 | 33 | |
| | BCLA01000003.1 (Scf3) | CEN3 | 220,773 | 221,345 | 573 | 27.9 | |
| | BCLA01000004.1 (Scf4) | CEN4 | 410,594 | 411,387 | 794 | 33.2 | |
| | BCLA01000005.1 (Scf5) | CEN5 | 524,594 | 525,105 | 512 | 24.8 | |
| | BCLA01000006.1 (Scf6) | CEN6 | 133,647 | 134,324 | 678 | 33.6 | |
| | BCLA01000007.1 (Scf7) | CEN7 | 408,363 | 409,067 | 705 | 34.2 | |
| | BCLA01000008.1 (Scf8) | CEN8 | 398,756 | 399,423 | 668 | 32.5 | |
| *M. vespertilionis* CBS15041 | KZ454987.1 (Scf1) | CEN1 | 410,820 | 411,340 | 521 | 15.7 | 56.6 |
| | KZ454988.1 (Scf2) | CEN2 | 1,275,509 | 1,276,238 | 730 | 25.8 | |
| | KZ454989.1 (Scf3) | CEN3 | 322,361 | 323,277 | 917 | 38.2 | |
| | KZ454990.1 (Scf4) | CEN4 | 583,450 | 584,319 | 870 | 28.9 | |
| | KZ454991.1 (Scf5) | CEN5 | 802,843 | 804,042 | 1200 | 28.8 | |
| | KZ454992.1 (Scf6) | CEN6 | 739,896 | 740,558 | 663 | 22.3 | |
| | KZ454993.1 (Scf7) | CEN7 | 268,699 | 269,626 | 928 | 28.8 | |
| | KZ454994.1 (Scf8) | CEN8 | 10,985 | 11,865 | 881 | 28 | |
| | KZ454995.1 (Scf9) | CEN9 | 19,047 | 19,724 | 678 | 29.1 | |
| *M. japonica* JCM11963 | BCKY01000001.1 (Scf1) | CEN1 | 1,068,050 | 1,068,614 | 564 | 25.1 | 62.35 |
| | BCKY01000002.1 (Scf2) | CEN2 | 139,423 | 139,920 | 497 | 20.3 | |
| | BCKY01000003.1 (Scf3) | CEN3 | 350,068 | 350,603 | 535 | 24.3 | |
| | BCKY01000004.1 (Scf4) | CEN4 | 380,877 | 381,439 | 562 | 24.2 | |
| | BCKY01000005.1 (Scf5) | CEN5 | 507,632 | 508,230 | 598 | 24.5 | |
| | BCKY01000006.1 (Scf6) | CEN6 | 240,968 | 250,550 | 582 | 23.7 | |
| | BCKY01000007.1 (Scf7) | CEN7 | 286,711 | 287,234 | 523 | 24.9 | |
| | BCKY01000008.1 (Scf8) | CEN8 | 87,314 | 87,873 | 559 | 23.9 | |
| | BCKY01000010.1 (Scf10) | CEN9 | 230,906 | 231,456 | 530 | 24.1 | |

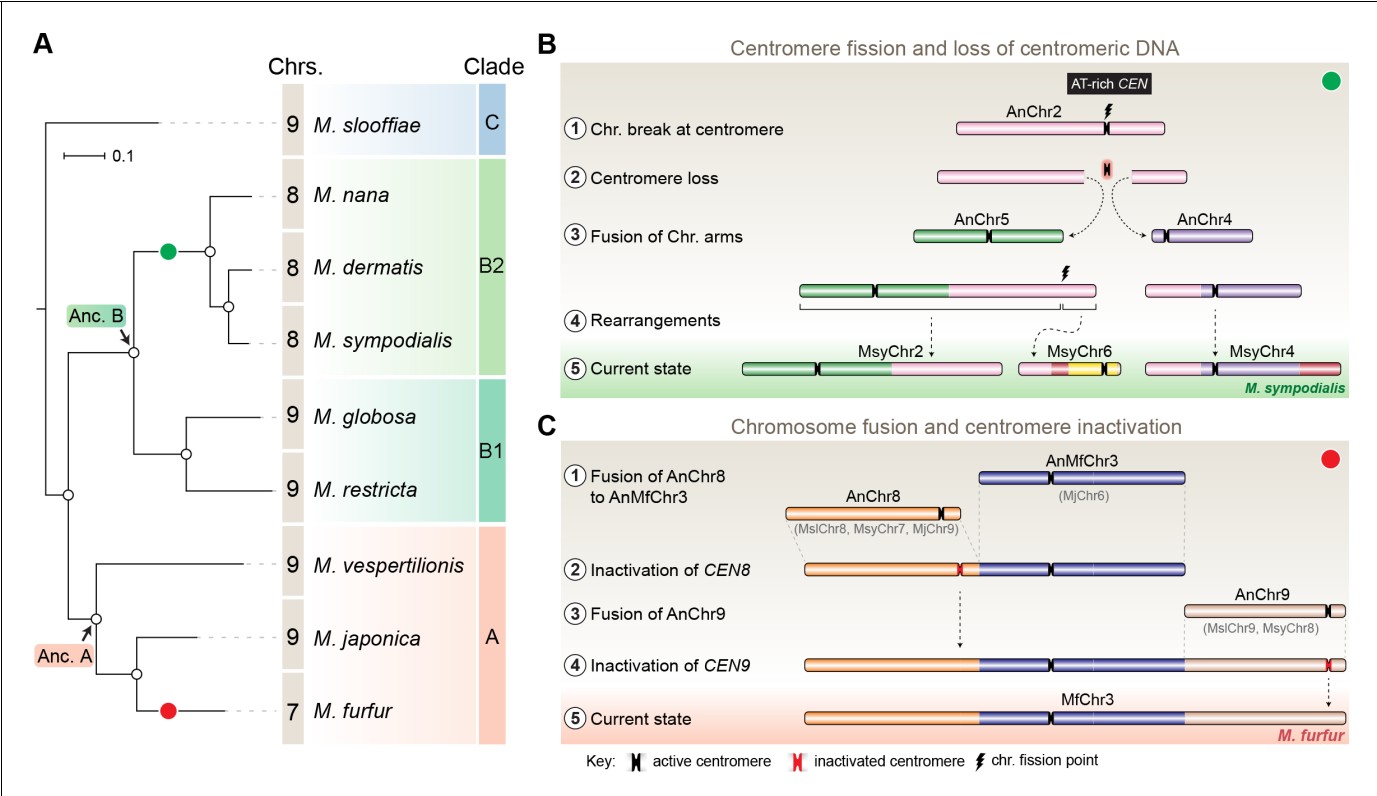

**Figure 7.** Karyotype evolution by loss of centromere function in *Malassezia* species. (**A**) Phylogenetic relationships between the *Malassezia* species analyzed in this study are represented and their chromosome numbers are shown. Species representing each clade are color-coded on the basis of previous reports (*Theelen et al., 2018*). The chromosome numbers for *M. slooffiae* and *M. globosa* are based on results from this study. In the case of *M. sympodialis*, *M. restricta*, and *M. furfur*, the chromosome numbers are based on previous reports (*Boekhout and Bosboom, 1994*; *Senczek et al., 1999*; *Zhu et al., 2017*). For *M. dermatis*, *M. nana*, *M. vespertilionis*, and *M. japonica*, the number of chromosomes were estimated from the predicted number of centromeres. The nodes corresponding to the ancestral state for Clade A and Clade B are labeled 'Anc. A' and 'Anc. B', respectively. Green and red circles indicate the origins of karyotypes that have eight and seven chromosomes, respectively, from an ancestral state of nine chromosomes. (**B**) Schematic of the centromere loss by breakage and the resulting reduction in chromosome number as observed in *M. sympodialis* (represented as the current state). A karyotype with nine chromosomes (as shown for *M. globosa*) is depicted as the ancestral state. (**C**) Proposed model of centromere inactivation observed in *M. furfur* as a consequence of fusion of AnChr8 and AnChr9 to the AnMfChr3 equivalent, resulting in a seven-chromosome configuration. The fusion product corresponding to extant MfChr3 is represented as the current state.

The online version of this article includes the following figure supplement(s) for figure 7:

**Figure supplement 1.** Putative centromeres of *M. dermatis*, *M. nana*, *M. vespertilionis*, and *M. japonica* map to global GC troughs in each chromosome.

**Figure supplement 2.** The 12-bp AT-rich motif is enriched at the putative centromeres of *M. dermatis, M. nana, M. vespertilionis,* and *M. japonica*.

**Figure supplement 3.** All of the predicted centromeres of *M. sympodialis, M. nana,* and *M. dermatis* belong to gene synteny blocks that are also conserved in species containing nine chromosomes.

**Figure supplement 4.** *MgCEN2* and *MgCEN3* are predicted to form secondary structures.

Based on the binding patterns of the kinetochore protein across the *M. sympodialis* genome, the 3–5-kb long region can be divided into two domains: (a) an AT-rich *CEN* core that maps to the intergenic region containing the GC trough, which shows maximum kinetochore binding (<1 kb); and (b) the regions flanking the core, which show basal levels of kinetochore protein binding. We observed conservation of the 12-bp AT-rich motif in the centromere core across the nine *Malassezia* species. It should also be noted that the 12-bp motif is significantly enriched at the centromeres but is not exclusive to the centromeres as it is detected across the chromosomes. We did not observe any orientation bias for these motifs. The functional significance of the frequent occurrence of this motif at centromeres is unknown. It will be intriguing to test the roles played by this motif, the core, and the flanking sequences in centromere function. Testing these domains for centromere function in vivo by making centromeric plasmids in various *Malassezia* species is challenging at present because of

technical limitations. Other than *M. sympodialis*, *M. pachydermatis,* and *M. furfur*, no other *Malassezia* species have been successfully transformed (*Ianiri et al., 2016*; *Celis et al., 2017*; *Ianiri et al., 2017a*). Moreover, in all of these cases, the genetic manipulations are performed by *Agrobacterium*-mediated transconjugation, which cannot be used for the introduction of circular plasmids. Hence, the functional significance of the 12-bp motif remains unknown and remains an important question to be addressed in the future to provide an understanding of centromere function in these species.

The centromeres in *M. sympodialis* contain transcribed ORFs, which have also been documented in the centromeres of rice, maize, and *Zymoseptoria tritici* (*Nagaki et al., 2004*; *Wang et al., 2014*). In contrast to these cases, our read count analysis did not reveal any significant difference in the transcription (RPKM values) of centromere-associated ORFs and ORFs elsewhere in the genome of *M. sympodialis*. We posit that the 12-bp AT-rich motif sequences could facilitate the transcription of these genes by recruiting transcription factors that have a possible role in kinetochore assembly. Binding of Cbf1 at CDEI in *S. cerevisiae* centromeres and of Ams2 at the central core sequences of *S. pombe* centromeres are classic examples of transcription factors facilitating kinetochore stability in fungal systems (*Hemmerich et al., 2000*; *Chen et al., 2003*). A fine regulation of transcription by Cbf1, Ste12, and Htz1, and the resulting low-level cenRNAs, have been implicated in proper chromosome segregation in budding yeast (*Ohkuni and Kitagawa, 2011*; *Ling and Yuen, 2019*). These studies reinforce the role of transcription in centromere function irrespective of the centromere structure.

In this study, we report three high-quality chromosome-level genome assemblies and identified centromeres in nine *Malassezia* species, representing all of the three *Malassezia* clades with differing numbers of chromosomes. This will serve as a rich resource for comparative genomics in the context of niche adaptation and speciation. Analysis of gene synteny conservation across centromeres using these genomes revealed breakage at the centromere as one of the mechanisms that results in a karyotype change between closely related species — those with nine chromosomes, such as *M. sloofiae* and *M. globosa*, and those with eight chromosomes, such as *M. sympodialis*. Gene synteny breakpoints adjacent to the centromeres have been reported in *C. tropicalis*, which has seven chromosomes, one less than *C. albicans* (*Chatterjee et al., 2016*). Centromere loss by breakage was proposed to have reduced the *Ashbya gossypii* karyotype by one when compared to the pre-whole genome duplication ancestor (*Gordon et al., 2011*). Breakpoints of conserved gene synteny between mammalian and chicken chromosomes were also mapped to the centromeres (*International Chicken Genome Sequencing Consortium, 2004*). Similar consequences in the karyotype have been reported in cases where centromeres were experimentally excised. Besides neocentromere formation, survival by fusion of acentric chromosome arms has been shown in *S. pombe* (*Ishii et al., 2008*). Such fusions are detected upon deletion of centromeres in another basidiomycete, *C. deuterogattii* (*Schotanus and Heitman, 2019*). By comparing the ancestral state (*M. sloofiae*) and other *Malassezia* species with either the same number or fewer chromosomes, we observed gene synteny breaks that were adjacent to centromeres (indicated by partial synteny conservation), in addition to the break observed at *MgCEN2* or *MslCEN5*. Does this suggest that *Malassezia* centromeres are fragile in nature? We advance the following hypothesis to explain the observed breaks at centromeres.

Studies of the common fragile sites in the human genome suggest different forms of replication stress as a major source of instability and subsequent breakage at these sites (*Helmrich et al., 2011*; *Letessier et al., 2011*; *Ozeri-Galai et al., 2011*). The resolution of the resulting replication fork stall has been shown to be critical for the stability of these fragile sites (*Schwartz et al., 2005*). Studies of the human fragile site FRA16D show that the AT-rich DNA (Flex1) results in fork stalling as a consequence of cruciform or secondary structure formation (*Zhang and Freudenreich, 2007*). Centromeres are natural replication fork stall sites in the genome (*Greenfeder and Newlon, 1992*; *Smith et al., 1995*; *Mitra et al., 2014*). The AT-rich core centromere sequence in *M. globosa* is also predicted to form secondary structures (*Figure 7—figure supplement 4*), which can be facilitated by the inherent replication fork stall at the centromeres. Whenever these secondary structures are unresolved and the fork restart fails, double strand breaks (DSBs) can occur at the centromeres. Chromosomal breakage and aneuploidy resulting from such defects are known to occur in cancers (*Kops et al., 2005*). In mammals, centromeric DSBs are repaired efficiently compared to regions elsewhere in the genome, largely because of the presence of several homology tracts in the form of repetitive DNA sequences and the stiffness provided by the inherent heterochromatic state, which

facilitates ligation (*Rief and Löbrich, 2002*). *Malassezia* species are haploid in nature and lack typical pericentric heterochromatin marks. Although the efficiency of centromeric DSB repair in the absence of long tracts of homologous sequences is not known in this species complex, we propose that the AT-rich core sequences, by virtue of secondary structure formation during DNA replication, could occasionally undergo DNA breakage at the centromere in *Malassezia* species.

The second mechanism of chromosome number reduction suggested by our analyses comparing the *M. furfur* genome with the genomes of *M. slooffiae* or *M. globosa* involves the inactivation of centromeres in the process of transition from a nine-chromosome state to a seven-chromosome state. Centromere inactivation occurs in cases involving the fusion of centric chromosomal fragments, stabilizing the fusion product and generating a functionally monocentric chromosome, as seen during the origin of human Chr2 from the ancestor shared with the great apes (*Yunis and Prakash, 1982*; *IJdo et al., 1991*). A larger proportion of known centromere inactivation events were shown to be mediated by epigenetic modifications, in which inactivated centromeres are enriched with marks such as H3K9me2/3, H3K27me2/3, or DNA methylation, emphasizing the role of heterochromatin in this process (*Zhang et al., 2010*; *Koo et al., 2011*; *Sato et al., 2012*). Deletion of the centromere sequence corresponding to kinetochore binding has also been reported as an alternate mechanism, albeit a less frequent mechanism, in both humans and in yeast (*Stimpson et al., 2010*; *Gordon et al., 2011*; *Sato et al., 2012*). In difference to what might be expected according to the two modes described above, we observed divergence in the sequences corresponding to the inactivated centromeres (*CEN8* and *CEN9* of both *M. slooffiae* and *M. globosa*) in the arms of *M. furfur* Chr3, resulting in the loss of AT-richness of these centromere core regions (*Figure 7C*). This is also suggestive of a functional role for AT-rich DNA in centromere function in these species.

A change in chromosome number between two closely related species such as *C. albicans* and *C. tropicalis* is associated with a change in centromere structure: centromeres in *C. albicans* are unique short regions that are epigenetically regulated, whereas those in *C. tropicalis* are associated with genetically defined homogenized inverted repeats (*Chatterjee et al., 2016*). Strikingly, in both the transitions described above for *Malassezia* species, we did not observe any change in the centromere structure. The emergence of evolutionarily new centromeres, as seen in primate evolution, was not detected in *Malassezia* species (*Rocchi et al., 2009*; *Kalitsis and Choo, 2012*). This is particularly striking in the absence of conservation of any specific centromere-exclusive DNA sequence. This suggests that a strong driving force helps to maintain the highly conserved centromere properties in closely related *Malassezia* species that descended from a common ancestor, even after extensive chromosomal rearrangements involving centromeres that might have driven speciation. Furthermore, centromere inactivation/loss of centromere function seems to be a conserved theme mediating variation in chromosome number from unicellular yeast species to metazoans, including primates.

## Materials and methods

The reagents, strains, plasmids, and primers used in this study are listed in the *Key Resources Table* below.

**Key resources table**

| Reagent type (species) or resource | Designation | Source or reference | Identifier | Additional information |
|---|---|---|---|---|
| Biological sample (*Malassezia globosa*) | CBS7966 | CBS | | Wild-type strain |
| Biological sample (*Malassezia slooffiae*) | CBS7956 | CBS | | Wild-type strain |
| Biological sample (*Malassezia furfur*) | CBS14141 | CBS | | Wild-type strain |
| Biological sample (*Malassezia sympodialis*) | ATCC42132 | ATCC | | Wild-type strain |

*Continued on next page*

*Continued*

| Reagent type (species) or resource | Designation | Source or reference | Identifier | Additional information |
|---|---|---|---|---|
| Biological sample (*Saccharomyces cerevisiae*) | BY4741 | | | Wild-type strain (*MATa his3Δ1 leu2Δ0 met15Δ0 ura3Δ0*) |
| Genetic reagent (*Malassezia sympodialis*) | MSY001 | This study | | *M. sympodialis* (ATCC42132) cells expressing GFP-Mtw1 |
| Genetic reagent (*Malassezia furfur*) | MF001 | This study | | *M. furfur* (CBS14141) cells expressing CENP-A-3xFLAG |
| Recombinant DNA reagent | Plasmid pGI3 | (*Ianiri et al., 2017b*) | | Vector backbone for *Agrobacterium*-mediated transconjugation |
| Recombinant DNA reagent | Plasmid pAIM1 | (*Ianiri et al., 2016*) | | Plasmid with *NAT* marker, used as template for PCR |
| Recombinant DNA reagent | Plasmid pVY7 | (*Kozubowski et al., 2013*) | | Plasmid with GFP, used as template for PCR |
| Recombinant DNA reagent | Plasmid pMHR04 | This study | | Construct used to tag MsyMtw1 with GFP tag at the amino-terminus |
| Recombinant DNA reagent | Plasmid pMF1 | This study | | Construct used to tag MfCENP-A with 3xFLAG tag at the carboxyl terminus |
| Antibody | Anti-GFP (mouse monoclonal) | Roche | 11814460001 | IF (1:100) WB (1:3000) |
| Antibody | Anti-PSTAIRE (mouse monoclonal) | Abcam | Cat. no. 10345 | WB (1:5000) |
| Antibody | Anti-mouse IgG HRP (goat polyclonal) | Abcam | Cat. no. ab97023 | WB (1:10,000) |
| Antibody | Anti-H3 (rabbit polyclonal) | Abcam | Cat. no. ab1791 | ChIP (5 μL per 500 μL IP fraction) |
| Antibody | Anti-H4 (rabbit polyclonal) | Abcam | Cat. no. ab10158 | ChIP (5 μL per 500 μL IP fraction) |
| Antibody | Anti-FLAG (M2) (mouse monoclonal) | Sigma | Cat. no. F1804 | WB (1: 2500) |
| Other | GFP-trap beads | ChromoTek | Cat. no. gta-20 | ChIP (20 μL per 500 μL fraction) |
| Other | Blocked agarose beads | ChromoTek | Cat. no. bab-20 | ChIP (20 μL per 500 μL fraction) |
| Other | Protein-A sepharose beads | Sigma | Cat. no. P3391 | ChIP (20 μL per 500 μL fraction) |
| Other | M2 anti-FLAG affinity gel | Sigma | Cat. no. A2220 | ChIP (20 μL per 500 μL fraction) |
| Other | Lysing enzymes from *Trichoderma harzianum* | Sigma | Cat. no. L1412 | Enzyme to prepare spheroplasts |
| Other | Zymolyase 20T | MP biomedicals | Cat. no. 320921 | Enzyme to prepare spheroplasts |
| Other | Chitosanase | Sigma | Cat. no. C0794 | Enzyme to prepare spheroplasts |
| Other | SensiFAST SYBR No ROX Kit | Bioline | Cat. no. BIO-98020 | qPCR assay reagent |
| Chemical compound, drug | 2-mercaptoethanol | HiMedia | Cat. no. MB041 | |
| Software, algorithm | Geneious 9.0 | Biomatters Ltd. | | |
| Software, algorithm | SyMap | (*Soderlund et al., 2011*) | | |

*Continued on next page*

*Continued*

| Reagent type (species) or resource | Designation | Source or reference | Identifier | Additional information |
|---|---|---|---|---|
| Software, algorithm | PhylloGibbs-MP | (*Siddharthan, 2008*) | | |
| Software, algorithm | Circos | (*Krzywinski et al., 2009*) | | |
| Software, algorithm | Satsuma | (*Grabherr et al., 2010*) | | |
| Software, algorithm | Easyfig | (*Sullivan et al., 2011*) | | |
| Software, algorithm | GET_HOMOLOGUES | (*Contreras-Moreira and Vinuesa, 2013*) | | |
| Software, algorithm | IQ-TREE v1.6.5 | (*Nguyen et al., 2015*) | | |
| Software, algorithm | iTOL v4.3.3 | (*Letunic and Bork, 2019*) | | |
| Software, algorithm | SMRTPortal v2.3 | PacBio, Menlo Park, CA, USA | | |
| Software, algorithm | Fiji | National Institutes of Health | | |
| Software, algorithm | Illustrator | Adobe Systems | | |
| Software, algorithm | Excel | Microsoft | | |
| Software, algorithm | Word | Microsoft | | |
| Sequence-based reagent | List of primers used in this study | Sigma | | In *Supplementary file 1* |

## Media and growth conditions for *Malassezia* strains

*Malassezia* strains were grown on modified Dixon's media (malt extract 36 g/L, desiccated ox-bile 20 g/L, Tween40 10 mL/L, peptone 6 g/L, glycerol 2 mL/L, oleic acid 2.89 mL/L). *M. sympodialis, M. furfur* strains were grown at 30°C. Cultures of *M. globosa* and *M. slooffiae* were grown at 32°C.

## Construction of the *M. sympodialis* strain expressing GFP-Mtw1

The allele for N-terminal tagging of Mtw1 with GFP was prepared by gap repair in the *Saccharomyces cerevisiae* BY4741 strain (*Eckert-Boulet et al., 2012*). Briefly, a 1.6-kb fragment consisting of the upstream and promoter sequence of the *MTW1* gene and a 1.6-kb fragment having the *MTW1* ORF (Protein ID: SHO76526) along with the downstream sequence were amplified from *M. sympodialis* genomic DNA. The GFP ORF (without the stop codon) and *NAT* were amplified from plasmids pVY7 and pAIM1, respectively. *S. cerevisiae* was transformed with all four fragments and the linearized plasmid pGI3 (digested with KpnI and BamHI) and the epitope-tagged allele were assembled in an ordered way by gap repair. Total DNA was isolated from *S. cerevisiae* and the *E. coli* DH5α strain was transformed. The pGFP-Mtw1 construct was screened by restriction digestion and further confirmed by sequencing. The pGFP-Mtw1 construct was used to transform *M. sympodialis* strain ATCC42132 by *Agrobacterium tumefaciens*-mediated transconjugation (*Ianiri et al., 2016*; *Ianiri et al., 2017a*).

## Construction of the *M. furfur* strain expressing CENP-A FLAG

The allele for C-terminal tagging of CENP-A with a 3xFLAG epitope tag was prepared by gap repair in the *Saccharomyces cerevisiae* BY4741 strain (*Eckert-Boulet et al., 2012*). Briefly, a 1-kb fragment consisting of the upstream and promoter sequence of the CENP-A gene of *M. furfur* including the ORF (CENP-A ORF coordinates in Chr1: 1,453,468–1,453,921), as well as a 1-kb fragment containing the sequence downstream of CENP-A ORF, were amplified from *M. furfur* genomic DNA. The 3xFLAG tag was introduced in the reverse primer annealing to the CENP-A ORF. The *NAT* marker

was amplified from plasmid pAIM1 as above. *S. cerevisiae* was transformed with all three fragments and plasmid pGI3 (digested with KpnI and BamHI) and the epitope-tagged allele were assembled in an ordered way by gap repair. Total DNA was isolated from *S. cerevisiae* and the *E. coli* DH5α strain was transformed. The resulting pMF1 construct was screened by restriction digestion and further confirmed by sequencing. The pMF1 construct was used to transform *M. furfur* strain CBS14141 by *Agrobacterium tumefaciens*-mediated transconjugation (*Ianiri et al., 2016*; *Ianiri et al., 2017a*) to obtain the epitope-tagged strain MF001.

## Microscopic imaging of live cells and processing

The GFP-Mtw1 strain was inoculated to 1% v/v from a saturated starter culture grown in mDixon medium. After growth for 6 hr at 30 ℃/ 150 rpm, these cells were pelleted at 4,000 rpm and washed three times with 1x phosphate-buffered saline (PBS), and the cell suspension was placed on a clean glass slide. A coverslip was placed on the spot and sealed prior to imaging. The images were acquired at room temperature using a laser scanning inverted confocal microscope LSM 880-Airyscan (ZEISS, Plan Apochromat 63x, NA oil 1.4) equipped with highly sensitive photo-detectors. The filters used were GFP/FITC 488 excitation and GFP/FITC 500/550 band pass, long pass for emission. Z-stack images were taken every 0.3 μm and processed using ZEISS Zen software or ImageJ. All of the images were processed post-acquisition with minimal adjustments to levels and linear contrast until the signals were highlighted.

## Preparation of spheroplasts

Cells were grown on mDixon's medium and washed with water by centrifugation at 4000 rpm for 5 min. Cells were resuspended in 10 mL of 5% (v/v) 2-mercaptoethanol solution in water and incubated at 30 ℃/ 150 rpm for 45 min. The cells were pelleted, washed, and resuspended in 3 mL spheroplasting buffer (40 mM citric acid, 120 mM $Na_2HPO_4$ and 1.2 M sorbitol) for every $1.5 \times 10^9$ cells. Cell clumps were dissociated by mild sonication for 30 s using the medium-intensity setting in a Bioruptor (Diagenode). Lysing enzymes from *Trichoderma harzianum* (Sigma), chitosanase (Sigma), and zymolyase-20T (MP Biomedicals) were added at 20 mg/mL, 0.2 μg/mL, and 100 μg/mL, respectively. The spheroplasting suspension was incubated at 30℃/65 rpm for 6–8 hr. The suspension was examined under a microscope to estimate the proportion of spheroplasts. Spheroplasts were washed with ice-cold 1xPBS and used as per the experimental design (adapted from *Boekhout, 2003*).

## Indirect immunofluorescence

The GFP-Mtw1 strain was inoculated to 1% (v/v) from a saturated starter culture grown in mDixon medium. After growth for 6 hr, the cells were fixed by the addition of formaldehyde to a final concentration of 3.7% for 1 hr. Post-fixing, the cells were washed with water and taken for preparation of spheroplasts (as described above). Spheroplasts were washed with ice-cold 1xPBS and resuspended in ice-cold 1xPBS to a cell density suitable for microscopy. Slides for microscopy were washed with water and coated with poly L-lysine (15 μL of 10 mg/mL solution per well) for 5 min at room temperature. The solution was aspirated and washed once with water. The cell suspension was added to each well (15–20 μL) and allowed to stand at room temperature for 5 min. The cell suspension was aspirated and the slides were washed once with water to remove unbound cells. The slides were fixed in ice-cold methanol for 6 min followed by treatment with ice-cold acetone for 30 s. Post fixing, blocking solution (2% non-fat skim milk in 1xPBS) was added to each well, and slides were incubated at room temperature for 30 min. After this, the blocking solution was aspirated and primary antibodies were added (mouse anti-GFP antibodies [Sigma] at 1:100 dilution). After incubation for 1 hr at room temperature, each slide was washed eight times with 1xPBS giving a 2 min incubation for every wash. Secondary antibody solution (goat anti-mouse-AlexaFluor488 [Invitrogen] at 1:500 dilution) was added to each well and incubated for 1 hr in the dark at room temperature. Post-incubation, slides were washed as described above. Mounting medium (DAPI at 100 ng/mL in 70% glycerol) was added, incubated for 5 min and aspirated out. Slides were sealed with a clean coverslip before imaging. The images were acquired at room temperature using an inverted fluorescence microscope (ZEISS Axio Observer, Plan Apochromat 100x, NA oil 1.4). Z- stack images were taken every 0.3 μm and processed using ZEISS Zen software/ImageJ.

## Chromatin immunoprecipitation (ChIP)

The ChIP protocol was adapted from that implemented for *C. neoformans* (*Yadav et al., 2018b*). Logarithmically grown cells were fixed with formaldehyde at a final concentration of 1% for 30 min for Mtw1 ChIP and 15 min for CENP-A, histone H3, and histone H4 ChIP. The reaction was quenched by the addition of glycine to a final concentration of 0.135 M. Cells were pelleted and processed for spheroplasting as described above. Spheroplasts were washed once sequentially using 10 mL of the following ice-cold buffers: 1xPBS, Buffer-I (0.25% Triton X-100, 10 mM EDTA, 0.5 mM EGTA, 10 mM Na-HEPES [pH 6.5]), and Buffer-II (200 mM NaCl, 1 mM EDTA, 0.5 mM EGTA, 10 mM Na-HEPES [pH 6.5]). The pellet after the final wash was resuspended in 1 mL lysis buffer (50 mM HEPES [pH 7.4], 1% Triton X-100, 140 mM NaCl, 0.1% Na-deoxycholate, 1 mM EDTA) for every $1.5 \times 10^9$ cells. Protease inhibitor cocktail was added to 1x final concentration.

The resuspended spheroplasts were sonicated with a Bioruptor (Diagenode) using a 30 s ON/OFF pulse at high-intensity mode with intermittent incubation on ice to obtain chromatin fragments in the size range of 100–300 bp. The lysate was cleared after sonication by centrifugation at 13,000 rpm for 10 min at 4°C. The input DNA fraction was separated at this step (1/10th volume of lysate) and processed for de-crosslinking by the addition of 400 μL elution buffer (0.1 M NaHCO$_3$, 1% SDS) per 100 μL lysate (processing for de-crosslinking is mentioned below). The remaining lysate was split equally and processed as IP and control samples. For GFP-Mtw1 ChIP, 20 μL GFP-trap beads and blocked agarose beads, respectively, were used for IP and control. For CENP-A-3xFLAG ChIP, 20 μL anti-FLAG affinity gel and blocked agarose beads, respectively, were used for IP and control. In the case of histone H3 or histone H4 ChIP, 5 μL of antibodies were used per IP fraction along with 20 μL Protein-A sepharose beads. Samples were rotated for 6 hr at 4°C. Post incubation, samples were sequentially washed as follows: twice with 1 mL low salt wash buffer (0.1% SDS, 1% Triton X-100, 2 mM EDTA, 20 mM Tris [pH 8.0], 150 mM NaCl), twice with 1 mL high salt wash buffer (0.1% SDS, 1% Triton X-100, 2 mM EDTA, 20 mM Tris [pH 8.0], 500 mM NaCl), once with 1 mL LiCl wash buffer (0.25 M LiCl, 1% NP-40, 1% Na-deoxycholate, 1 mM EDTA, 10 mM Tris [pH 8.0]) and twice with 1 mL 1xTE (10 mM Tris [pH 8.0], 1 mM EDTA). Samples were rotated in a rotaspin for 5 min at room temperature for every wash (15 min in case of histone H3 or histone H4 ChIP). After washing, DNA was eluted from the beads twice using 250 μL elution buffer. The samples for elution were incubated at 65°C for 5 min, rotated for 15 min and then collected by centrifugation.

Samples were decrosslinked by the addition of 20 μL 5 M NaCl and incubation at 65°C for 6 hr. Following this, samples were deproteinized by the addition of 10 μL 0.5 M EDTA, 20 μL 1 M Tris [pH6.8], 2 μL Proteinase K (20 mg/L) and incubation at 45°C for 2 hr. After incubation, samples were treated with an equal volume of phenol-chloroform-isoamyl alcohol (25:24:1) mix, and the aqueous phase was extracted by centrifugation. DNA was precipitated by the addition of 1/10th volume of 3 M Na-acetate, 1 μL glycogen (20 mg/mL), 1 mL absolute ethanol and incubation at −20°C for at least 12 hr. Finally, the samples were harvested by centrifugation at 13,000 rpm for 45 min at 4°C followed by washing the pellet once with ice-cold 70% ethanol. Air-dried pellets were then resuspended in 20 μL sterile MilliQ water with 10 μg/mL RNAse. Samples were either processed for library preparation for ChIP-sequencing or analyzed by qPCR with the primers listed in *Supplementary file 1*.

## Analysis of sequencing data

GFP-Mtw1 ChIP sequencing was performed at the Clevergene Biocorp. Pvt. Ltd., Bengaluru, India. A total of 46,704,720 and 63,524,912 150-bp paired-end reads were obtained for IP and Input samples, respectively. The reads were mapped to the *M. sympodialis* ATCC42132 genome using Geneious 9.0 (http://www.geneious.com/) with default conditions. Each read was allowed to map only once randomly anywhere in the genome. The alignments were exported to BAM files, sorted, and visualized using the Integrative Genomics Viewer (IGV, Broad Institute). The images from IGV were imported into Adobe Photoshop (Adobe) and scaled for representation purposes. RNA-sequencing data (E-MTAB-4589) from a previous study was downloaded from the ArrayExpress website, sorted, and visualized using IGV. GC-content was calculated using Geneious 9.0 with a sliding window size of 250 bp. The data were exported as wig files and further visualized using IGV.

## Western blotting

Protein lysates for western blot were prepared by the TCA method. Overnight grown cultures of 1 mL were harvested, washed, and resuspended in 400 µL of 12.5% ice-cold TCA solution. The suspension was vortexed briefly and stored at – 20°C for 4–6 hr. The suspension was thawed on ice, pelleted at 14,000 rpm for 10 min, and washed twice with 350 µL of 80% acetone (ice-cold). The washed pellets were air-dried completely and resuspended in the desired volume of lysis buffer (0.1 N NaOH+1% SDS). Samples were separated in 12% polyacrylamide gels and transferred onto nitrocellulose membranes. For probing, mouse anti-GFP antibody (Roche) and the HRP conjugated goat anti-mouse secondary antibody (Abcam) were used at 1:3000 and 1:5000 dilution, respectively, in 2.5% skim milk powder in 1xPBS. The blots were developed using Chemiluminescence Ultra substrate (BioRad) and imaged using the VersaDoc imaging system (BioRad).

## PFGE analysis for *M. globosa* and *M. slooffiae*

For CHEF analysis of *M. globosa* (CBS7966) and *M. slooffiae* (CBS7956), the cells were grown on solid mDixon medium and then collected and resuspended in 1xPBS. CHEF plugs were prepared as described in previous studies (*Sun et al., 2014*; *Sun et al., 2017*). Chromosomes were separated in 1% Megabase certified agarose gels made with 0.5 × TBE, using a BioRad CHEF-DR II System running at 3.2 V/cm with linear ramping switching time from 90 to 360 s for 120 hr in 0.5xTBE at 14°C. The gel was stained with EtBr and visualized under UV.

For the chromoblot analyses of *M. globosa* (CBS7966), the gel from the CHEF analysis was first transferred to a membrane, and the resulting chromoblots were then hybridized sequentially with four probes from chromosomes 3, 4, 5, and 6 of the CBS7966 genome assembly, respectively (see the *Supplementary file 1* for the primer information), as described in previous studies (*Findley et al., 2012*; *Yadav et al., 2018b*).

## *M. globosa* genome assembly

Sequence reads were assembled with HGAP3 included in SMRTPortal v2.3 (PacBio, Menlo Park, CA, USA) and default parameters, except for the genome size set to 9 Mb. Assembly completeness was evaluated by checking for telomeric repeats. Non-telomeric contig-ends were aligned to other contigs using BLAST and unique overlaps were used to build complete chromosomes. Short telomere ends were extended using uniquely mapped reads longer than 10 kb and repolishing of the assembly using the resequencing pipeline in SMRTPortal v2.3. The assembly resulted in 19 contigs, with a total length of 9.2 Mb. 17 long and one short telomere could be identified. Six contigs had telomeres on both the 5'- and the 3'-end, and thus represent full-length chromosomes (Chr1, 2, 3, 6, 7, and 8). Six contigs had only one telomere and seven contigs had no telomeric sequence. Two contigs without telomeres were from the mitochondrion and two were from the ribosomal repeats. Chromosome 5 was constructed from two contigs ending in ribosomal rDNA repeats. The assembly contains six copies of the repeat, but read coverage suggests a length of 30–40 repeat units that cannot be resolved with the available read-length. The remaining contigs were used to build chromosomes 4 and 9, which share highly similar 5'-ends. The two ends can be distinguished by two microsatellite expansions. Chromosome 4 had a very short 3'-telomere from the default assembly, but the raw data contained a uniquely mapping read that extended several repeat units past the assembly end. After polishing the reference, all nine chromosomes had clear 5'- and 3'- telomeres.

## *M. slooffiae* genome assembly

Sequence reads were assembled using HGAP3 included in SMRTPortal v2.3 (PacBio, Menlo Park, CA, USA) with default parameters. This resulted in an assembly with 14 scaffolds with telomeric repeats at both ends in nine contigs. Of the remaining five contigs, three of them could be assigned to mitochondrial DNA on the basis of BLAST analysis with *M. globosa*. The remaining two contigs, of size 5.8 kb and 2.3 kb, did not show BLAST hits against the *M. globosa* or *M. sympodialis* genomes.

## *M. furfur* genome assembly

Sequence reads generated from CBS14141 were assembled using HGAP3 included in SMRTPortal v2.3 (PacBio, Menlo Park, CA, USA) with default parameters. This resulted in an assembly with eight

scaffolds, of which the nuclear genome was organized in seven scaffolds that had telomere repeats at both ends.

## Gene synteny analysis

Analysis of gene synteny conservation across the centromeres was performed by BLAST as follows. The genomes for *M. restricta*, *M. nana*, and *M. dermatis* were downloaded from the NCBI genomes portal. The PacBio assembled genomes of *M. globosa* and *M. slooffiae* were used for synteny analysis. Synteny analysis was done in the context of ORFs flanking the centromeres of *M. sympodialis*. The protein sequences for each of these ORFs served as the query in BLAST analysis against the genome of other species. The local database for each genome was set up in the Geneious software for this analysis. The percentage identity values for each of the ORFs are mentioned in the boxes in *Figure 7—figure supplement 3*. In addition, synteny analyses between *M. globosa* and *M. sympodialis* were conducted with megablast (word size: 28) and plotted together with GC content (calculated as the deviation from the genomic mean, in non-overlapping 1-kb windows), using Circos (v0.69–6) (*Krzywinski et al., 2009*). Additional whole-genome alignments were conducted with Satsuma (*Grabherr et al., 2010*), with default parameters. The linear synteny comparisons shown in *Figures 5* and *6* were generated with the Python application EasyFig (*Sullivan et al., 2011*).

## Species phylogeny

To reconstruct the phylogenetic relationship among the nine *Malassezia* species selected, orthologs were identified using the bidirectional best-hit (BDBH), COGtriangles (v2.1), and OrthoMCL (v1.4) algorithms implemented in the GET_HOMOLOGUES software package (*Contreras-Moreira and Vinuesa, 2013*). The proteome of *M. sympodialis* ATCC42132 was used as a reference. A phylogeny was inferred from a set of 738 protein sequences as follows. Individual protein sequences were aligned using MAFFT v7.310 (L-INS-i strategy), and poorly aligned regions were trimmed with TrimAl (-gappyout). The resulting alignments were concatenated to obtain a final supermatrix consisting of a total of 441,200 amino acid sites (159,270 parsimony-informative). This sequence was input to IQ-TREE v1.6.5 (*Nguyen et al., 2015*) and a maximum likelihood phylogeny was estimated using the LG+F+R4 amino acid model of substitution. Branch support values were obtained from 10,000 replicates of both ultrafast bootstrap approximation (UFBoot) and the nonparametric variant of the approximate likelihood ratio test (SH-aLRT) implemented in IQ-TREE. The best likelihood tree was graphically visualized with iTOL v4.3.3 (*Letunic and Bork, 2019*).

## Acknowledgements

We thank Clevergene Biocorp Pvt. Ltd., Bengaluru for generating the Mtw1 ChIP-sequencing data. SRS is a Senior Research Fellow supported by intramural funding from JNCASR. MHR was a National Postdoctoral Fellow (PDF/2016/002858), supported by the Science and Engineering Research Board (SERB), Department of Science and Technology (DST), Government of India. KS is a Tata Innovation Fellow (grant number BT/HRT/35/01/03/2017) and is supported by a grant for Life Science Research, Education and Training (BT/INF/22/SP27679/2018) of Department of Biotechnology, Government of India and intramural funding from JNCASR. Studies were supported in part by NIH/NIAID R37 award AI39115-21 and R01 award AI50113-15 to JH. JH is co-director and fellow of the CIFAR program Fungal Kingdom: Threats and Opportunities. TLD acknowledges the A* STAR Industry alignment fund H18/01a0/016, Asian Skin Microbiome Program. We thank the members of KS lab and JH lab for valuable discussions and comments during bi-weekly Skype meetings. We thank the referees and the editors for their constructive suggestions that helped us to improve this study.

## Additional information

### Funding

| Funder | Grant reference number | Author |
| --- | --- | --- |
| Department of Biotechnology , Ministry of Science and Technology | Tata Innovation Fellowship BT/HRT/35/01/03/2017 | Kaustuv Sanyal |

| Department of Biotechnology, Ministry of Science and Technology | BT/INF/22/SP27679/2018 | Kaustuv Sanyal |
|---|---|---|
| National Institutes of Health | R37 award-AI39115-21 | Joseph Heitman |
| Agency for Science, Technology and Research | H18/01a0/016 | Thomas L Dawson Jr |
| Jawaharlal Nehru Centre for Advanced Scientific Research | Graduate student fellowship | Sundar Ram Sankaranarayanan |
| Science and Engineering Research Board | PDF/2016/002858 | Md Hashim Reza |
| National Institutes of Health | R01 award-AI50113-15 | Joseph Heitman |
| Jawaharlal Nehru Centre for Advanced Scientific Research | | Kaustuv Sanyal |

The funders had no role in study design, data collection and interpretation, or the decision to submit the work for publication.

### Author contributions

Sundar Ram Sankaranarayanan, Conceptualization, Data curation, Formal analysis, Investigation, Methodology, Writing - original draft, Writing - review and editing, performed western blot, microscopy, synteny conservation analysis, ChIP-qPCR analysis, and prepared the samples for ChIP-sequencing; Giuseppe Ianiri, Formal analysis, Investigation, Methodology, Writing - review and editing, Transformation of *Malassezia* strains and PCR confirmation; Marco A Coelho, Data curation, Formal analysis, Validation, Investigation, Visualization, Methodology, Writing - review and editing, performed genome comparisons, phylogenetic analysis, and prepared the figures; Md Hashim Reza, Formal analysis, Investigation, Writing - original draft, Writing - review and editing, generated the constructs for tagging MsyMtw1 and MfCENP-A, and synteny conservation analysis; Bhagya C Thimmappa, Formal analysis, performed synteny conservation analysis; Promit Ganguly, Formal analysis, analyzed the ChIP-seq data and RNA-seq data for *M. sympodialis*; Rakesh Netha Vadnala, Formal analysis, Visualization, performed GC/GC3 analysis; Sheng Sun, Formal analysis, Investigation, Writing - review and editing, performed PFGE and chromoblot analysis of *M. slooffiae* and *M. globosa*; Rahul Siddharthan, Software, Formal analysis, Visualization, Writing - review and editing, identified the 12 bp motif and performed motif scan analysis across *Malassezia* genomes; Christian Tellgren-Roth, Resources, Methodology, genome assemblies of *M. globosa*; Thomas L Dawson Jnr, Resources, Writing - review and editing, genome assemblies of *M. globosa*, *M. slooffiae*, and *M. furfur*; Joseph Heitman, Conceptualization, Resources, Supervision, Funding acquisition, Project administration, Writing - review and editing; Kaustuv Sanyal, Conceptualization, Resources, Supervision, Funding acquisition, Writing - original draft, Project administration, Writing - review and editing

### Author ORCIDs

Giuseppe Ianiri ![ORCID] https://orcid.org/0000-0002-3278-8678
Rahul Siddharthan ![ORCID] https://orcid.org/0000-0002-2233-0954
Kaustuv Sanyal ![ORCID] https://orcid.org/0000-0002-6611-4073

### Decision letter and Author response

Decision letter https://doi.org/10.7554/eLife.53944.sa1
Author response https://doi.org/10.7554/eLife.53944.sa2

# Additional files

### Supplementary files

• Supplementary file 1. This file contains the names, sequences, and description of primers used in this study.

• Transparent reporting form

## Data availability

The Mtw1 ChIP sequencing reads reported in this paper have been deposited under NCBI BioProject (Accession number PRJNA509412). The genome sequence assemblies of *M. globosa*, *M. slooffiae*, and *M. furfur* have been deposited in GenBank with accession numbers SAMN10720087, SAMN10720088, and SAMN13341476 respectively.

The following datasets were generated:

| Author(s) | Year | Dataset title | Dataset URL | Database and Identifier |
|---|---|---|---|---|
| Sankaranarayanan SR, Ianiri G, Coelho MA, Reza MH, Thimmappa BC, Ganguly P, Vadnala RN, Sun S, Siddharthan R | 2019 | Genome assembly of *Malassezia slooffiae* | https://www.ncbi.nlm.nih.gov/biosample/10720088 | NCBI BioSample, SAMN10720088 |
| Sankaranarayanan SR, Ianiri G, Coelho MA, Reza MH, Thimmappa BC, Ganguly P, Vadnala RN, Sun S, Siddharthan R | 2019 | Genome assembly of *Malassezia globosa* | https://www.ncbi.nlm.nih.gov/biosample/SAMN10720087 | NCBI BioSample, SAMN10720087 |
| Sankaranarayanan SR, Ianiri G, Coelho MA, Reza MH, Thimmappa BC, Ganguly P, Vadnala RN, Sun S, Siddharthan R | 2019 | Genome assembly of *Malassezia furfur* | https://www.ncbi.nlm.nih.gov/biosample/SAMN13341476 | NCBI BioSample, SAMN13341476 |
| Sankaranarayanan SR, Ianiri G, Coelho MA, Reza MH, Thimmappa BC, Ganguly P, Vadnala RN, Sun S, Siddharthan R | 2020 | Identification of centromeres in *Malassezia sympodialis* | https://www.ncbi.nlm.nih.gov/bioproject/PRJNA509412/ | NCBI BioProject, PRJNA509412 |

The following previously published datasets were used:

| Author(s) | Year | Dataset title | Dataset URL | Database and Identifier |
|---|---|---|---|---|
| L'Oreal, Stanislas Morand | 2018 | *Malassezia restricta* CBS 7877 genome, complete sequence | https://www.ncbi.nlm.nih.gov/genome/702?genome_assembly_id=413940 | NCBI Genome, 413940 |
| Zhu Y, Engström PG, Tellgren-Roth C, Baudo CD, Kennell JC, Sun S, Billmyre RB, Schröder MS, Andersson A, Holm T, Sigurgeirsson B, Wu G, Sankaranarayanan SR, Siddharthan R, Sanyal K, Lundeberg J, Nystedt B, Boekhout T, Dawson TL Jr, Heitman J, Scheynius A, Lehtiö J | 2017 | Genome sequencing and integrative gene annotation of *Malassezia sympodialis* | https://www.ncbi.nlm.nih.gov/bioproject/PRJEB13283 | NCBI BioProject, PRJEB13283 |
| RIKEN Center for Life Science Technologies, Division of Genomic Technologies | 2016 | Genome sequencing of *Malassezia nana* JCM 12085 | https://www.ncbi.nlm.nih.gov/bioproject/313886 | NCBI BioProject, PRJDB3735 |

| RIKEN Center for Life Science Technologies, Division of Genomic Technologies | 2016 | Genome sequencing of *Malassezia dermatis* JCM 11348 | https://www.ncbi.nlm.nih.gov/bioproject/PRJDB3732 | NCBI BioProject, PRJDB3732 |
| --- | --- | --- | --- | --- |
| RIKEN Center for Life Science Technologies, Division of Genomic Technologies | 2016 | Genome sequencing of *Malassezia japonica* JCM 11963 | https://www.ncbi.nlm.nih.gov/bioproject/PRJDB3733 | NCBI BioProject, PRJDB3733 |
| Lorch JM, Palmer JM, Vanderwolf KJ, Schmidt KZ, Verant ML, Weller TJ, Blehert DS | 2017 | *Malassezia vespertilionis* strain: NWHC:44797-103 Genome sequencing and assembly | https://www.ncbi.nlm.nih.gov/bioproject/PRJNA393681 | NCBI BioProject, PRJNA393681 |

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
