## [Decision Letter]

**Acceptance summary:**

In this manuscript the authors identify the centromeres of several Malassezia species to trace the events that led to karyotype development in this species complex. The authors mapped centromeres in species with 8 or 9 chromosomes by Mtw1 ChIP. They discovered a 12 bp AT-rich consensus that is unique for centromeres but whose functional significance remains to be determined. They show convincingly that the reduction of chromosome number was accompanied by a break of one chromosome at its centromere and subsequent fusion of the chromosome arms with other chromosomes to reduce the number of chromosomes in the karyotype by one.

**Decision letter after peer review:**

[Editors’ note: the authors submitted for reconsideration following the decision after peer review. What follows is the decision letter after the first round of review.]

Thank you for submitting your work entitled "Centromere-mediated chromosome break drives karyotype evolution in closely related *Malassezia* species" for consideration by *eLife*. Your article has been reviewed by three peer reviewers, including Wolf-Dietrich Heyer as the Reviewing Editor and Reviewer #1, and the evaluation has been overseen by a Senior Editor.

Our decision has been reached after consultation between the reviewers. Based on these discussions and the individual reviews below, we regret to inform you that your work will not be considered further for publication in *eLife*.

As you can see from the reviews, there was a span of opinions, which converged at the end of our discussion. We all agreed that the manuscript is interesting and that the topic of karyotype evolution is important. If you elect to address the comments in a future submission to *eLife*, I would be happy to serve as a reviewing editor again.

*Reviewer #1:*

In this manuscript the authors identify the centromeres of several *Malassezia* species to identify the events that led to karyotype development in this species complex. The authors mapped centromeres in species with 8 or 9 chromosomes by Mtw1 ChIP and identified a 12 bp AT-rich consensus that is unique for centromeres. They show convincingly that the reduction of chromosome number was accompanied by a break of one chromosome at its centromere and subsequent fusion of the chromosome arms with other chromosomes to reduce the number of chromosomes in the karyotype by one.

1) The classic molecular marker for the centromere in CENP-A or CENP-C enrichment. The depletion of normal histone H3 is used here as an argument that the identified regions are centromeres, but the more classic criterion would be better.

As there is little doubt that the identified regions are centromeres, I am unsure, whether this experiment must be in the revision or whether it is simply a possible suggested addition.

*Reviewer #2:*

Changes of chromosome numbers are related to centromere inactivation events and/or chromosome fusions. Because karyotype changes seem to be related to speciation, it is important to characterize and compare centromere organization at genome level between closed species. In this paper, authors focused on Malassezia species, who belong to Basiodiomycota and have variable karyotypes. They experimentally defined 8 centromeres in Malassezia sympodialis by ChIP-seq with kinetochore protein Mtw1 and proposed a 12 bp consensus motif in centromeres. Furthermore, they examined synteny and predicted centromeres in other Malassezia species and found that some centromeres are located in a synteny breakpoint. Finally they proposed a model how karyotype diversity occurred through centromere breakage and inactivation during evolution.

Overall, subject itself is interesting and it important to understand karyotype evolution. However, to improve quality of the paper, some additional experiments are needed. My major concern is lack of functional analysis of the 12 bp sequence. Identification of this sequence is one of highlights in this paper, but there was no functional assay. Authors should present functional significance of this sequence. In addition, they also should confirm centromere sequence using ChIP experiments in other Malassezia species.

*Reviewer #3:*

This manuscript describes the identification of centromeres in Malassezia sympodialis using ChIP-seq of a GFP-tagged kinetochore protein (Mtw1). Centromeres are described as being 3-5 kb with an AT-rich core enriched with a 12 bp consensus motif. Using these features, the authors predict centromere locations in the genomes of several other Malassezia species, including *M. globosa, M. slooffiae* and *M. restricta*. These three species have 9 chromosomes, whilst *M. sympodialis* has 8. Mapping of synteny breaks between the genomes leads the authors to suggest that centromere loss has resulted in *M. sympodialis* having one less chromosome than the other species. The authors speculate that chromosome breakage due to merotelic centromere attachments drive karyotype diversity in *Malessezia*. The genomes of *M. globosa* and *M. slooffiae* are also assembled de novo.

The manuscript contains potentially interesting data on Malassezia centromeres and karyotype diversity. However, the merotely model presented is highly speculative and overplayed.

Merotely requires that a centromere/kinetochore is capable of binding at least two microtubules (in merotely a single kinetochore is bound to microtubules emanating from both poles). Point centromeres which bind only one microtubule are not capable of merotelic attachments. Large regional centromeres such as vertebrate centromeres bind multiple microtubules per centromere, as do the regional centromeres of *S. pombe* (2-4 microtubules bound per kinetochore). Smaller regional centromeres, such as those of *Candida albicans* (3-5 kb) have been reported to bind only one microtubule (e.g. Joglekar et al., 2008, Burrack et al., 2011). Thus, it is entirely possible, given the size of the centromeres in Malessezia (3-5 kb – or even smaller – see below) that each binds only one microtubule. If that were the case, merotelic attachments would not be possible. Is there any data available in these species to suggest that the kinetochores bind multiple microtubules? Have lagging chromosomes (typical of merotely) been observed in these species during anaphase? There is no discussion of these points in the manuscript.

Could other processes have led to rearrangements involving ancestral *CEN2*? Perhaps centromeres are for some reason fragile (i.e. due to AT-richness), leading to 'ordinary' rearrangements like those on chromosome arms, without invoking centromere-specific processes such as merotely?

*Malesezzia* centromeres are described as being 3-5 kb, but it looks as if the Mtw1 ChIP-seq reads cover approximately 1 kb. It is very hard to interpret the data because of the low resolution in Supplementary Figure 1A and the lack of scale in Supplementary Figure 1B. Zooming in further on the ChIP-seq and providing a diagram of the positions of features such as 12 bp motif and% GC would be helpful and informative.

Figure 3—figure supplement 2C and Supplementary Figure 1F. No indication of size/region covered. Figure 3—figure supplement 2 appears to show that there are several 12 bp motifs in a 500 bp window at centromeres, but these are not apparent in Figure 3—figure supplement 2C and Supplementary Figure 1F.

Some means of confirming centromere location in the other Malassezia species is needed. Whilst the difficulty in transforming different yeast species and therefore tagging proteins is understood, have the authors contemplated raising an antibody to Cse4/CENP-A or testing for cross-reactivity of an existing antibody? Failing that, other Malassezia centromeres would be expected to have a reduced level of histone H3 incorporated, where it is replaced by CENP-A, compared to chromosome arms (as in Supplementary Figure 1D).

Figure 2B: a negative control locus for GFP-Mtw1 should be included.

[Editors’ note: further revisions were suggested prior to acceptance, as described below.]

Thank you for resubmitting your work entitled "Loss of centromere function drives karyotype evolution in closely related Malassezia species" for further consideration by *eLife*. Your revised article has been reviewed by three peer reviewers, one of whom is a member of our Board of Reviewing Editors, and the evaluation has been overseen by Kevin Struhl as the Senior Editor.

The manuscript has been improved but there a few remaining issues that need to be addressed before acceptance, as outlined below:

Essential revisions:

1) The identification of the 12 bp AT-rich consensus that is unique for centromeres may suggest functional importance, but this is currently not experimentally shown. The manuscript should clearly state that the importance of the 12 bp consensus sequence is not known.

Reviewer #2:

Authors of this manuscript have submitted a previous manuscript on karyotype in Malassezia species, which was rejected by *eLife* ~9 months ago. Here, they added various new data and submit a new manuscript. Changes of chromosome numbers might be associated with centromere inactivation or centromere brake events followed by chromosome fusions. Although comparative genome analyses in close species suggest such possibilities, it is not well characterized in very closed species. Because karyotype changes seem to be related to speciation, it is important to characterize and compare centromere organization at genome level between closed species. In this manuscript, authors focused on Malassezia species and characterized whole genome sequences and centromere positions of several Malassezia species, which have different numbers of chromosomes. Based on comparison of genome synteny and chromosome numbers, authors conclude that ancestral Malassezia species has 9 chromosomes, but chromosome breakage events or centromere inactivation events independently occur in different lineages, and these events associated with chromosome fusion caused reduction of chromosome numbers in some of Malassezia species.

I think that the manuscript is substantially revised and data themselves have high quality. However, as I asked in previous review, it is important to clarify the 12 bp sequence for centromere function. It is not clear whether the centromere is specified by sequence dependent or independent. In their model, the centromere is not created in speciation in Malassezia after centromere loss. In general, after centromere loss, chromosome can fuse with another chromosome or acentric chromosome can get a neocentromere. As this species does not form new centromere, it is really important to address whether the centromere is specified by sequence dependent or not. Although it might be hard to do genetic manipulation in this species, it might be possible to deplete the 12 bp motif. If they can show the significance of the 12bp sequence in any methods, paper would be suitable for *eLife*.

In addition, they evaluated CENP-A accumulation as reduction of H3. Quantitative ChIP-seq is not easy. If they want to do that, they should normalize seq-reads using the Spike-in method.

---

## [Author Response]

[Editors’ note: the authors resubmitted a revised version of the paper for consideration. What follows is the authors’ response to the first round of review.]Reviewer #1:[…]1) The classic molecular marker for the centromere in CENP-A or CENP-C enrichment. The depletion of normal histone H3 is used here as an argument that the identified regions are centromeres, but the more classic criterion would be better.As there is little doubt that the identified regions are centromeres, I am unsure, whether this experiment must be in the revision or whether it is simply a possible suggested addition.

We thank reviewer 1 for the careful review and the positive feedback on the data set. We concur with the reviewer 1’s comment that CENP-A and CENP-C are the classical markers for centromeres. The proteins of the Mis12 complex are evolutionarily conserved across phyla and are shown to be enriched at the centromeres in various organisms (Goshima et al., 1999; Westerman et al., 2003; Goshima et al., 2003; Roy et al., 2011) and hence can be used as an authentic centromere marker. Also, the functionality of an epitope tagged CENP-A in a haploid organism remains uncertain. Taking these facts into consideration, we instead tagged the Mis12 homolog, Mtw1, with GFP to identify centromeres in *M. sympodialis*.

In centromeric nucleosomes, histone H3 is replaced by CENP-A. Reduced levels of histone H3 at the centromeres has been shown in *Candida lusitaniae*, in which centromeres are similar in length (Kapoor et al., 2015) to those observed in *M. sympodialis*. To confirm centromere identity of Mtw1-enriched regions in the *M. sympodialis* genome, we demonstrated reduced levels of histone H3 enrichment as it is expected that histone H3 molecules are replaced by centromere-specific histone H3 variant CENP-A. This further supports that the Mtw1-bound regions are indeed centromeres in *M. sympodialis*. This has been mentioned in the revised manuscript (Introduction: eighth paragraph; Results: subsection “Histone H3 is depleted at the core centromere with active genes at the pericentric regions in *M. sympodialis*”).

Reviewer #2:[…]Overall, subject itself is interesting and it important to understand karyotype evolution. However, to improve quality of the paper, some additional experiments are needed. My major concern is lack of functional analysis of the 12 bp sequence. Identification of this sequence is one of highlights in this paper, but there was no functional assay. Authors should present functional significance of this sequence.

We appreciate that the reviewer finds this study, and karyotype evolution in general, important and relevant. Functional analysis of the 12 bp motif identified by our study is challenged by technical limitations described below. Among the *Malassezia* species, only *M. sympodialis, M. furfur,* and *M. pachydermatis* have been successfully transformed in the laboratory. In all three cases, transformation was accomplished with *Agrobacterium* mediated trans-conjugation which is not amenable for episomal maintenance of plasmids. Any sequence provided within the T-DNA border elements is integrated into the genome upon transconjugation. This precludes us from performing functional assays such as stabilization of a plasmid containing the centromere or the associated 12 bp motif. This has been described in the Discussion section in the revised manuscript (third paragraph).

In addition, they also should confirm centromere sequence using ChIP experiments in other Malassezia species.

Following the reviewer’s recommendation, we have experimentally validated the centromeres in three other species: *M. furfur, M. globosa,* and *M. slooffiae*. These results are presented in Figure 4 of the revised manuscript (Results: subsection “Centromeres in *M. furfur*, M. *slooffiae*, and *M. globosa* map to chromosomal GC minima”).

1) For *M. furfur,* we generated strains functionally expressing CENP-A with a 3xFLAG at the carboxy terminus. All seven centromeres we predicted in *M. furfur* were validated by ChIP-qPCR using this strain (Figure 4B and C).

2) In the case of *M. slooffiae* and *M. globosa*, we used the reduction in histone H3 levels (as observed in *M. sympodialis*) to validate centromere identity. Histone H4 was used as a control for the presence of nucleosomes in both cases. The reduction in the relative abundance of H3/H4 at the centromeres as compared to a control region away from and unlinked to the centromere is included (Figure 4D, E). Furthermore, the enrichment profile of H3/H4 across one centromere and the flanking pericentric region in both *M.*

*slooffiae* and *M. globosa* has also been included (Figure 4F and G).

Reviewer #3:This manuscript describes the identification of centromeres in Malassezia sympodialis using ChIP-seq of a GFP-tagged kinetochore protein (Mtw1). Centromeres are described as being 3-5 kb with an AT-rich core enriched with a 12 bp consensus motif. Using these features, the authors predict centromere locations in the genomes of several other Malassezia species, including M. globosa, M. slooffiae and M. restricta. These three species have 9 chromosomes, whilst M. sympodialis has 8. Mapping of synteny breaks between the genomes leads the authors to suggest that centromere loss has resulted in M. sympodialis having one less chromosome than the other species. The authors speculate that chromosome breakage due to merotelic centromere attachments drive karyotype diversity in Malessezia. The genomes of M. globosa and M. slooffiae are also assembled de novo.The manuscript contains potentially interesting data on Malassezia centromeres and karyotype diversity. However, the merotely model presented is highly speculative and overplayed.Merotely requires that a centromere/kinetochore is capable of binding at least two microtubules (in merotely a single kinetochore is bound to microtubules emanating from both poles). Point centromeres which bind only one microtubule are not capable of merotelic attachments. Large regional centromeres such as vertebrate centromeres bind multiple microtubules per centromere, as do the regional centromeres of *S. pombe* (2-4 microtubules bound per kinetochore). Smaller regional centromeres, such as those of Candida albicans (3-5 kb) have been reported to bind only one microtubule (e.g. Joglekar et al., 2008, Burrack et al., 2011). Thus, it is entirely possible, given the size of the centromeres in Malessezia (3-5 kb – or even smaller – see below) that each binds only one microtubule. If that were the case, merotelic attachments would not be possible. Is there any data available in these species to suggest that the kinetochores bind multiple microtubules? Have lagging chromosomes (typical of merotely) been observed in these species during anaphase? There is no discussion of these points in the manuscript.Could other processes have led to rearrangements involving ancestral CEN2? Perhaps centromeres are for some reason fragile (i.e. due to AT-richness), leading to 'ordinary' rearrangements like those on chromosome arms, without invoking centromere-specific processes such as merotely?

We thank the reviewer for the comprehensive review and raising this point and giving us the opportunity to improve our hypothesis. As mentioned by the reviewer, organisms with similar sized small regional centromeres (ranging from 3 to 5 kb) have been reported to have only one microtubule attachment per kinetochore, making merotelic attachments less likely in *Malassezia* species. Further, we do not have any direct evidence for merotelic attachments in any of these organisms. In the revised manuscript, we propose that the AT-rich centromeres could be fragile sites in the genome. Drawing parallels from our understanding of the instability caused by fragile sites in the human genome, we advance the following hypothesis. Instability of chromosomal fragile sites has been correlated with replication stress that results in replication fork stalling. Stable inheritance of these regions is directly related to the ability of cells to resolve the fork stall with the aid of DNA repair machinery. Centromeres are reported to be natural replication fork stall sites in several yeasts (Smith et al., 1995; Mitra et al., 2014). The presence of AT-rich or repetitive sequences at the fork stall sites can result in the formation of secondary structures. Such structures are usually repaired by the DNA repair machinery (Schwartz et al., 2005). However, failure of efficient repair may result in a double-strand break at the centromere. The outcome of repair of these broken ends depends on the availability of a homologous sequence. Absence of any DNA sequence homology can result in events such as translocations that we observed in *Malassezia* species. This is a likely possibility in the *Malassezia* species given the haploid nature of these organisms. We have discussed this model in the revised manuscript and removed the merotely model (Discussion section, sixth paragraph).

Malesezzia centromeres are described as being 3-5 kb, but it looks as if the Mtw1 ChIP-seq reads cover approximately 1 kb. It is very hard to interpret the data because of the low resolution in Supplementary Figure 1A and the lack of scale in Supplementary Figure 1B. Zooming in further on the ChIP-seq and providing a diagram of the positions of features such as 12 bp motif and% GC would be helpful and informative.

We thank the reviewer for pointing this out. The resolution of figure has been improved and this panel has been included in Figure 3 in the revised manuscript.

Figure 3—figure supplement 2C and Supplementary Figure 1F. No indication of size/region covered. Figure 3—figure supplement 2 appears to show that there are several 12 bp motifs in a 500 bp window at centromeres, but these are not apparent in Figure 3—figure supplement 2C and Supplementary Figure 1F.

For the dot-plot analysis, were used sequences of 5 kb in length flanking the Mtw1 enriched regions in *M. sympodialis*. We have included the coordinates of the sequences used for the dot-plot in the figure legend. We thank the reviewer for pointing this out. The motif we identified was developed from a position weight matrix. Hence the motifs represented in Figure 3—figure supplement 2 may not be identical in sequence thereby not being picked up in the dot-plot.

Some means of confirming centromere location in the other Malassezia species is needed. Whilst the difficulty in transforming different yeast species and therefore tagging proteins is understood, have the authors contemplated raising an antibody to Cse4/CENP-A or testing for cross-reactivity of an existing antibody? Failing that, other Malassezia centromeres would be expected to have a reduced level of histone H3 incorporated, where it is replaced by CENP-A, compared to chromosome arms (as in Supplementary Figure 1D).

We appreciate the reviewer’s understanding of the experimental limitations related to

*Malassezia*. As the reviewer suggested, we attempted to raise antibodies against the epitope ‘EANYASSASA’ in *M. furfur* CENP-A (75-84 aa), which we expected would cross-react with *M. globosa* (EENYDSSASA) and *M. sympodialis* (EANYDSSASA) CENP-A. The purified antisera could detect the peptide in ELISA at a titer of 1:64,000. However, we could not detect a protein corresponding to CENP-A by western blot at any of the dilutions tested (ranging between 1:200 and 1:2000). The results are included here (see Author response image 1).

Considering these results, we analyzed the depletion of histone H3 as an alternative approach to validate our centromere predictions in the two species with 9 chromosomes: *M. slooffiae* and *M. globosa*. In addition, we also generated *M. furfur* strains that functionally express CENP-A with a 3xFLAG tag at the carboxy terminus. We used this strain to experimentally to conduct ChIP-qPCR analysis of the candidate centromere regions, and thereby validate that the approaches employed for centromere prediction in *M. furfur* and related species are accurate. This has been further described in our response to a point raised by reviewer 2.

**Author response image 1. respfig1:** Figure corresponding to the generation of antibodies against CENP-A in *Malassezia* species. (**A**) A line diagram of MfCENP-A indicating the peptide sequence (EANYASSASA) at positions 75-84 aa used for generation of antibodies in rabbit. (**B**) Detection of the peptide epitope by antibodies before and after purification by ELISA. For this assay, 50 ng peptide was coated in each well and probed with varying dilutions of the antisera (1:4000, 1:32000, 1:64000). Absorbance at 450 nm was recorded after the addition of stopping solution and absorbance values are plotted for each sample in the *y-*axis. NC- no antigen control, (**C–E**) whole cell extracts from *M. globosa (Mg*, CBS7966 strain), *M. sympodialis (Msy*, ATCC42132 strain) and *M. furfur (Mf*, CBS14141 strain) were probed with pre-immune sera, purified CENP-A antisera, and histone H3 antisera in various dilutions as indicated. Lane L denotes the protein molecular weight markers. Both pre-immune sera and the purified antibodies were tested at a dilution of 1:200 while histone H3 antibodies were used at 1:2500 dilution.

Figure 2B: a negative control locus for GFP-Mtw1 should be included.

We would like to point out that the graph was plotted after normalizing with a non-centromeric locus. This has been mentioned in the revised axis label and legend (Figure 3—figure supplement 1B in the revised manuscript).

[Editors’ note: what follows is the authors’ response to the second round of review.]

Essential revisions:1) The identification of the 12 bp AT-rich consensus that is unique for centromeres may suggest functional importance, but this is currently not experimentally shown. The manuscript should clearly state that the importance of the 12 bp consensus sequence is not known.

We have revised the manuscript, as requested, to clearly state in the Discussion section (third paragraph) that for technical reasons, the function of the motif remains unknown. However, we would like to point out that this has already been mentioned in the aforementioned paragraph.

Reviewer #2:[…]I think that the manuscript is substantially revised and data themselves have high quality. However, as I asked in previous review, it is important to clarify the 12 bp sequence for centromere function. It is not clear whether the centromere is specified by sequence dependent or independent. In their model, the centromere is not created in speciation in Malassezia after centromere loss. In general, after centromere loss, chromosome can fuse with another chromosome or acentric chromosome can get a neocentromere. As this species does not form new centromere, it is really important to address whether the centromere is specified by sequence dependent or not. Although it might be hard to do genetic manipulation in this species, it might be possible to deplete the 12 bp motif. If they can show the significance of the 12bp sequence in any methods, paper would be suitable for eLife.In addition, they evaluated CENP-A accumulation as reduction of H3. Quantitative ChIP-seq is not easy. If they want to do that, they should normalize Seq-reads using the Spike-in method.

We concur with the reviewer that it is important to understand the nature of centromere function (sequence dependent or sequence independent) and the role of the 12 bp motif. At present, we are unable to ascribe a function to this motif due to the technical limitations described in the manuscript (Discussion, third paragraph) that we hope to address in our future studies.